



# El Niño Southern Oscillation signal in a new East Antarctic ice core, Mount Brown South

Camilla K. Crockart[1], Tessa R. Vance[1], Alexander D. Fraser[1], Nerilie J. Abram[2], Alison S. Criscitiello[3], Mark A.J. Curran[4,1], Vincent Favier[5], Ailie J.E. Gallant[6], Helle A. Kjær[7], Andrew R. Klekociuk[4,1], Lenneke M. Jong[4,1], Andrew D. Moy[4,1], Christopher T. Plummer[1], Paul T. Vallelonga[7,#], Jonathon Wille[5], and Lingwei Zhang[8]

[1]Australian Antarctic Program Partnership, Institute for Marine & Antarctic Studies, University of Tasmania, Hobart 7004, Australia.
[2]Research School of Earth Sciences and ARC Centre of Excellence for Climate Extremes, Australian National University, Canberra ACT 2601, Australia.
[3]Department of Earth and Atmospheric Sciences, University of Alberta, Edmonton, T6G 2R3, Canada.
[4]Australian Antarctic Division, Channel Highway, Kingston 7050, Australia.
[5]Institut des Géosciences de l'Environnement, Université Grenoble-Alpes, Grenoble, France.
[6]School of Earth, Atmosphere and Environment, Monash University, Rainforest Walk, Clayton 3800, Victoria.
[7]Physics of Ice, Climate and Earth, Niels Bohr Institute, University of Copenhagen, Denmark.
[8]Institute for Marine & Antarctic Studies, University of Tasmania, Hobart 7004, Australia.

*Correspondence to*: Camilla K. Crockart (camilla.crockart@utas.edu.au)

[#]Now at UWA Oceans Institute, University of Western Australia, Perth 6909, Australia.

## Abstract

Paleoclimate archives, such as high-resolution ice core records, provide a means to investigate long-term (multi-centennial) climate variability. Until recently, the Law Dome (Dome Summit South) ice core record remained one of few long-term high-resolution records in East Antarctica. A new ice core drilled in 2017/2018 at Mount Brown South, approximately 1000 km west of Law Dome, provides an additional high-resolution record that will likely span the last millennium in the Indian Ocean sector of East Antarctica. Here, we compare snowfall accumulation rates and sea salt concentrations in the upper portion (~21 m) of the Mount Brown South record, and an updated Law Dome record over the period 1975-2016. Annual sea salt concentrations from the Mount Brown South record preserves a stronger signal for the El Niño-Southern Oscillation (ENSO; in austral winter and spring, $r = 0.521$, $p < 0.000$, Niño 3.4) compared to the Law Dome record (November-February, $r = -0.387$, $p = 0.018$, Niño 3.4). The Mount Brown South and Law Dome ice cores record inverse signals for the ENSO, suggesting the occurrence of distinct moisture and aerosol intrusions. We suggest that ENSO-related sea surface temperature anomalies in the equatorial Pacific drive atmospheric teleconnections in the southern mid-latitudes. These anomalies are associated with a weakening (strengthening) of regional westerly winds to the north of Mount Brown South that corresponds to years of low (high) sea salt deposition at Mount Brown South during La Niña (El Niño) events. The Mount Brown South annual sea salt





record when complete will offer a new proxy record for reconstructions of the ENSO over the recent millennium, along with

improved understanding of regional atmospheric variability in the southern Indian Ocean in addition to that derived from Law Dome.

## 1.0 Introduction

Ice cores collected from the Antarctic ice sheet contain chemical signals that are used to reconstruct past climate conditions. Ice cores can be categorised as low-resolution (millennially to centennially-resolved) and high-resolution (seasonally to

annually-resolved) records. Low-resolution ice cores collected from low-accumulation zones of the Antarctic Plateau contain climate signals dating back hundreds of thousands of years. High-resolution ice cores contain more detailed climate signals but cover shorter timescales. These are drilled in coastal regions in Antarctica, where snowfall accumulation is relatively high and basal complexity is minimal (Vance et al. 2016). Only a few regions in Antarctica have sufficient annual snowfall accumulation rates to allow for long-term (millennial timescale) high-resolution ice cores (Vance et al. 2016). These include

James Ross Island near the Antarctic Peninsula (57.411° W, 64.121° S; Abram et al. 2011; Mulvaney et al. 2012), the West Antarctic Ice Sheet (WAIS) Divide in West Antarctica (79.467° S, 112.085° W; Bisiaux et al. 2012), the Roosevelt Island (Climate Evolution; RICE) in West Antarctica (79.218° S, 161.423° W; Winstrup et al. 2019), and Law Dome (LD) in East Antarctica (66.461° S, 112.841° E; Morgan et al. 1997; van Ommen et al. 2004; Roberts et al. 2015). Despite comprising a large proportion of the continental landmass, East Antarctica has a particularly poor spatial coverage of long-term high-

resolution ice cores (Vance et al. 2016). Multiple studies have highlighted the need for additional high-resolution ice cores in the Indian Ocean sector of East Antarctica (Stenni et al. 2017; Stocker et al. 2013; Thomas et al. 2017; Vance et al. 2016). These high-resolution records are required to fill spatial gaps in reconstructions of Antarctic temperature variability, aid in calibrating radar estimates of net surface mass balance, and provide additional proxy records that enhance the confidence and reliability of global climate reconstructions. Additional ice cores may also contain signals for major sources of climate

variability over the past millennia, including the dominant modes of climate variability (MOCV) in the Southern Hemisphere; the Southern Annular Mode (SAM), the El Niño-Southern Oscillation (ENSO), and the Indian Ocean Dipole (IOD). Understanding how these MOCV have varied in the past is important for a global initiative, the Past Global Changes 2k Network (PAGES 2k), that aims to integrate regional climate proxies to create global climate reconstructions over the past two millennia (PAGES 2k Consortium 2013). Expanding the PAGES 2k Network enables climate modelers to better understand

natural climate variability, which in turn may generate more reliable projections of future climate.

The dominant MOCV in the southern high-latitudes is the SAM, characterised by synchronous atmospheric anomalies of opposing signs in the high- and mid-latitudes (Marshall 2003). These entail low (high) pressure anomalies associated with a strengthening (weakening) of the circumpolar vortex and a contraction (expansion) of the westerly wind belt during positive

(negative) SAM phases. SAM variability lacks the persistence of coupled ocean-atmosphere MOCV, so can change phase on





weekly to seasonal timescales. A number of multi-centennial reconstructions of past SAM variability from tree-ring and high-resolution ice core records have been developed (e.g., Abram et al. 2014; Datwyler et al. 2017; Villalba et al. 2012; Zhang et al. 2010). Abram et al. (2014) developed a reconstruction of SAM variability (1000-2007 CE) using 25 proxy records, including an oxygen isotope record from the James Ross Island ice core. The lack of a SAM proxy in the high-latitudes of the

Indian Ocean means that these reconstructions do not represent the full circumpolar SAM state (Hessl et al. 2017). Goodwin et al. (2004) developed a proxy of wind strength in the south-western Pacific Ocean related to the SAM variability in sea salt deposition (in early austral winter) at LD. However, more recent work by Marshall et al. (2017) suggests that LD preserves only a weak signal of the SAM toward the negative phase in snowfall accumulation. Vance et al. (2016) suggest that a new ice core collected from west of LD may contain an independent SAM signal.


Although SAM is the dominant MOCV in the southern high-latitudes, the ENSO also influences the Antarctic climate. Globally, the ENSO is the primary interannual MOCV, and is characterised by opposing sea surface temperature anomalies (SSTA) in the western and eastern equatorial Pacific, and occurs at an inter-annual timescale (McPhaden et al. 2006; Dätwyler et al. 2019). Warm (cool) SSTA in the eastern equatorial Pacific are caused by the weakening (strengthening) of the easterly

trade winds during El Niño (La Niña) events (Markgraf and Diaz 2000; Trenberth 1997). Despite its origins in the equatorial Pacific, the ENSO influences the southern extra-tropical and Antarctic climate via teleconnections (linked atmospheric anomalies in two geographically separate regions) generated by Rossby Wave Trains (RWT; Ding et al. 2011; Turner et al. 2004). RWT result from reduced (enhanced) Walker Cell convection in the tropics in austral winter during El Niño (La Niña) events. This causes the Pacific South American (PSA) pattern of high-low-high (low-high-low) tropospheric pressure

anomalies that propagate south-eastward from the equatorial Pacific toward the South Pacific (Yiu and Maycock 2019). Proxy records for the ENSO from tree-ring, coral and ice core records have been used to produce multi-centennial reconstructions of the ENSO (e.g., Braganza et al. 2009; Carré et al. 2014; Cobb et al. 2003; Cobb et al. 2013; Dätwyler et al. 2019; Emile-Geay et al. 2016; Freund et al. 2019; Fowler et al. 2012; Grothe et al. 2019; Stahle et al. 1998). However, these reconstructions frequently disagree on past ENSO variability because both the individual proxies and the ENSO events that they respond to

are geographically separate and distinct, and the magnitude and persistence of the ENSO teleconnections vary between events (Braganza et al. 2009). There are numerous signals for the ENSO from West Antarctic ice core records due to their location at the southern end of the PSA pressure pattern (Turner et al. 2017). The PSA pattern is associated with strong atmospheric anomalies (e.g., warming near the Antarctic Peninsula) during austral winter, and to a lesser extent spring (Ding et al. 2011). El Niño (La Niña) events are also known to weaken (strengthen) the Amundsen Sea Low, a quasi-stationary low pressure

region in the South Pacific, and have a significant influence on the West Antarctic climate (Hosking et al. 2013). However, several studies also suggest a complex and limited relationship with ENSO, at least for the Surface Mass Balance (SMB) and melt, as part of summer variability is inherited from the previous austral winter (e.g., Donat-Magnin et al. 2020). Signals for the ENSO in East Antarctica are more muted, however a significant signal for the ENSO is preserved in the summer sea salt record from LD (Vance et al. 2013).




The IOD may also influence southern high-latitude climate, although the mechanism of this link is debated. The IOD is characterised by opposing SSTA in the western and eastern equatorial Indian Ocean, and occurs at an inter-annual scale (Saji et al. 1999). Cool (warm) SSTA in the eastern Indian Ocean during the positive (negative) IOD phase are associated with enhanced (reduced) ocean upwelling offshore of Sumatra and Java. Enhanced (reduced) coastal upwelling is caused by

strengthening (weakening) of the south-easterly trade winds. Positive IOD events result in increased Walker cell convection in the western tropical Indian Ocean, and decreased convection over Indonesia, which acts to sustain the SSTA and surface wind patterns (Webster et al. 1999). The IOD events begin in austral autumn, peak in spring and decay with the onset of the Australian monsoon, which typically occurs in early December (Cai et al. 2013). Although the IOD predominantly influences the climate in regions near the Indian Ocean basin, Abram et al. (2020) suggest that IOD-related convection anomalies cause

RWT patterns (similar to the PSA pressure pattern in the Pacific) that extend across the Australian continent towards the South Pacific. This suggests that the IOD may influence the southern mid-latitudes, which may in turn affect the Antarctic climate. Unlike the SAM and ENSO, it is unclear how the IOD interacts with the SAM in mid-latitudes, or whether the recent positive trend in the SAM enhances or suppresses any IOD link to the high-latitudes (Favier et al. 2016). Determining this will assist with understanding whether East Antarctic ice core signals could potentially be proxy reservoirs for the IOD, as reconstructions

for the IOD are currently limited, with only one semi-continuous reconstruction of the IOD variability over the recent millennium from coral records in the eastern equatorial Indian Ocean (Abram et al. 2020).

In this study, we present the first analysis of climate signals contained in the upper section of the new Mount Brown South ice core record (MBS, 69.111° S, 86.312° E), and an updated LD (Dome Summit South site, 66.461° S, 112.841° E) ice core

record. The MBS site was chosen according to a set of specific selection criteria by Vance et al. (2016): 1. 1,000-2,000-year-old ice at 300 m depth, but with sub-annual resolution using standard ice core analytical practices (e.g. at least 0.20 m yr$^{-1}$ ice equivalent (IE) accumulation). 2. Minimal surface reworking (estimated using a MODIS-based surface roughness estimate ground-truthed with laser altimetry flight line data). 3. Minimal ice displacement or elevation change at 300 m depth using Lagrangian streamline tracing and published ice velocities. 4. A significant teleconnection with the mid-latitudes distinct from

the Law Dome region (for detail of the above site selection criteria, see Vance et al., 2016). The MBS record is intended to be complementary to the LD record by preserving sub-decadal signals for climate variability from the western sector of the southern Indian Ocean. Previous short firn ice cores (5-10 m depth) have been collected from the coast inland to the Mount Brown region (Vance et al. 2016). Previous studies using these short ice cores have focused on proxies for local temperature using oxygen isotopes (e.g., Smith et al. 2002), and regional sea-ice extent using methanesulphonic acid (e.g., Foster et al.

2006). There are no studies using MBS ice cores to investigate potential signals for the ENSO, SAM, and IOD. The new MBS record presented here provides an opportunity to search for MOCV signals, which can later be integrated with established proxies to enhance and expand the confidence and reliability of past climate reconstructions, and aid in understanding regional atmospheric circulation variability. The new MBS record is unique in that it contains three short ice cores (20-25 m) that span





the full satellite era (1979-2017) in addition to the Main core, which is likely to span the past millennium to 2009. This coverage
of the satellite era will allow for a rigorous analysis of the variability of climate signals at the MBS site and dating uncertainties
associated with surface reworking. Here, we carry out detailed analyses of the annual snowfall accumulation rates and sea salt
concentrations in the MBS ice core record to assess the hypotheses that it; *(Hypothesis 1) contains signals for past climate
variability at a high-resolution, and (Hypothesis 2) contains climate signals that differ from the LD record.*

## 2.0 Methods

**2.1 Mount Brown South and Law Dome (Dome Summit South site)**

MBS (elevation 2,084 m) is a coastal wet-deposition site located in Wilhelm II Land in the Indian Ocean sector of East
Antarctica, approximately 1,000 km west of LD and 380 km east of Davis station (see Fig. 1). Net surface mass balance
estimates for the MBS region from airborne radar surveys suggest that the isochrone dating back 1,000 years lies at
approximately 300 m depth (Vance et al. 2016). Four ice cores were drilled at MBS in the summer of 2017/18 within 100 m
from one another. The drilling site included surface features of drifts and sastrugi that were up to 0.5 m high running in a
roughly easterly direction (see Appendix C for aerial photograph and details). The cores drilled include one long core,
MBS1718 (hereafter termed "Main", depth 295 m, Hans Tausen drill; Johnsen et al. 2007; Sheldon et al. 2014), and three short
cores, MBS1718-Alpha, MBS1718-Charlie, MBS1718-Bravo (hereafter "Alpha", "Bravo", and "Charlie", 20-25 m depth,
Kovacs drill). The Bravo core was collected exclusively for persistent organic pollutant analysis so it will not be considered
hereafter, although high resolution water stable isotope analyses from this core were considered for confirmatory purposes
during dating the records presented here. The MBS record likely spans the past millennium and deeper ice core analyses are
ongoing. However, precise dating has only been undertaken on the upper section (1975-2016) of the three cores considered
here. This period includes the satellite era, where satellite and reanalysis datasets are used to investigate the environmental
conditions associated with climate signals preserved in the ice cores.


LD (elevation 1,370 m, 66.461° S, 112.841° E) is also a wet-deposition coastal site located on a small coastal ice cap in Wilkes
Land in East Antarctica (see Fig. 1). The Dome Summit South (DSS) core drilling location is 4.6 km south of the LD summit,
has prevailing south-easterly winds, minimal katabatic influence, and preserves strong maritime signals (McMorrow et al.
2001; Plummer et al. 2012). Previous studies have reported snowfall accumulation of 0.680 m yr$^{-1}$ IE (Roberts et al. 2015; van
Ommen et al. 2004) and low mean wind speeds of 8.3 m s$^{-1}$ (Morgan and van Ommen 1997). The LD ice core used in this
study, DSS1617, was drilled in summer 2016/17 and is intended to replace the upper portion of the LD record presented in
Vance et al. (2013), which used four successive short ice cores to cover the period 1989-2009. The new record presented here
covers the period 1989-2016. This increases the overlap with the satellite era (and therefore calibration period) by 7 years (or
19%) and reduces dating errors associated with combining multiple ice core records. Prior to 1989, the LD ice core record
used in this study is the same as that used in Vance et al. (2013), DSS97.





## 2.2 Isotope and trace ion chemistry analyses

Discrete samples for water stable isotope and trace chemistry samples were cut under trace clean conditions. In order to investigate the optimal sample resolution over the satellite era for accurate dating, we cut 1.5 cm isotope samples in contrast
to 3 cm chemistry samples over the upper portion of the ice cores (top 40 m). For detailed information on the trace clean discrete analysis technique, see Plummer et al. (2012) and references therein. The oxygen isotopes and ion chemistry in the MBS and LD ice cores were analysed according to established methods (Curran and Palmer 2001; Curran et al. 2003; Palmer et al. 2001; Plummer et al. 2012). The VG Isogas SIRA mass spectrometer was used to determine the ratio of oxygen isotopes ($\delta^{18}O$) in the LD ice core, while the Picarro L2130-i isotopic water analyser was used to determine the water stable isotopes
ratios ($\delta^{18}O$ and dD) in the MBS ice cores. Isotopic values are expressed as per mil (‰) relative to the Vienna Standard Mean Oceanic Water (VSMOW) standard. The standard deviations of $\delta^{18}O$ for repeated measurements of laboratory reference water samples were less than 0.07 ‰ (for the LD ice core) and 0.5 ‰ (for the MBS ice cores). The Thermo-Fisher/Dionex ICS3000 ion chromatograph was used to determine the concentrations of trace ion chemistry (anions and cations), including sea salt concentrations (chloride ($Cl^-$), sodium ($Na^+$), magnesium ($Mg^{2+}$)), calcium ($Ca^{2+}$) and sulphate ($SO_4^{2-}$), as well as
methanesulfonic acid (MSA). Non sea salt sulphate ($nssSO_4^{2-}$) was calculated according to the methods in Plummer et al. (2012) and used in this study as a dating aid in addition to the usual suite of seasonal species above. Parallel sections of the MBS Main and Charlie cores were sent to the University of Copenhagen for impurity determination by Continuous Flow Analysis (Bigler et al., 2011), however we only consider the initial data available from the discrete analyses performed in Australia in this study.

## 2.3 Dating


Annual depth layers were assigned to the MBS and LD ice cores using seasonally varying species, principally $\delta^{18}O$, $nssSO_4^{2-}$, $Na^+$ and the ratio of $SO_4^{2-}/Cl^-$, which has been shown to be an excellent summer marker in the LD ice cores (Plummer et al., 2012). Similar to Morgan and van Ommen (1997), the summer maxima in $\delta^{18}O$ were fixed as January 8 (± 2 weeks). The climatology of hydrogen peroxide ($H_2O_2$) is also used in selected sections of the LD cores for confirmatory dating (e.g. van
Ommen & Morgan, 1996). Volcanic ash layers (indicated by $nssSO_4^{2-}$ peaks) linked to the Pinatubo volcanic eruption in the Philippines in mid-1991 were used as a reference depth horizon to cross-check the annual depth layer chronology (Plummer et al. 2012; see Fig. 1). Each ice core was dated individually and independently, without reference to other site records to ensure independence of the method.

## 2.4 Snowfall accumulation rates and sea salt concentration analyses

The annual depth layers were used in combination with an empirical density model (see Eq. 1) to determine the annual ice equivalent snowfall accumulation rates (hereafter accumulation) for the MBS ice cores. The empirical density model uses the



mid-point depth of the annual layer (d), the density of ice ($\rho$ = 0.917 g cm$^{-3}$), takes vertical strain thinning into account, and is based on the methods of Roberts et al. (2015). While allowing for strain thinning in firn is generally unnecessary, we wish our results to be directly comparable to the longer MBS accumulation records that are in development, hence the use of equation

200  1.

Empirical Density = [$\rho$] – [883.5356 * exp(-0.011078644) * d] + [436.8285] – [1.887488 * d]                Eq. (1)

Chloride was used as a proxy for sea salt in this study to maintain comparability with Vance et al. (2013). We acknowledge that Na$^+$ is the usual proxy for sea salt (particularly in the Northern Hemisphere) however, we wished to compare like-for-like with LD in Vance et al., 2013. The trace chemistry analysis in the Australian Antarctic Program Partnership ice core

laboratories is optimised to analyse anions (e.g., Cl$^-$) in order to derive highly accurate methanesulfonic acid (MSA), Cl$^-$, SO$_4^{2-}$ and nitrate (NO$_3$) records. The Na$^+$ and Cl$^-$ data streams are essentially indistinguishable ($r = 0.940$, $p < 0.000$), however we tend to have more data gaps in the Na$^+$ record because of this anion priority. We repeated our correlation analyses described below using Na$^+$ for the MBS record, with virtually identical results (see Appendix A). Although chloride is affected by post-depositional processes, and has sources other than sea salt (e.g., HCl), MBS has sufficient accumulation (Benassai et al. 2005)

to preserve the chloride record, similar to LD.

Sea salt concentrations (Cl$^-$, hereafter sea salts) were log-transformed to create a normally distributed record, as the raw concentrations are skewed toward infrequent high concentration events. The sea salts were then linearly interpolated (i.e., binned) to monthly resolution to determine the climatology of the sea salts in the MBS and LD record. Annual sea salt concentrations in the MBS ice cores were used for all other analyses to avoid errors associated with linear interpolation that

assume uniform accumulation throughout the year as seasonal variability in snowfall accumulation at MBS has not been investigated at this stage. The annual sea salts in the three MBS ice cores were averaged to minimise errors associated with dating uncertainties, such as timing noise (i.e., summer peaking analytes used as dating horizons depend on snowfall at the correct time to be preserved accurately, meaning that annual averages may include a small portion of signals into the adjacent year). The sea salt concentration for the year 1987 in the MBS Charlie ice core was excluded from this study (see Fig. 4). It

was difficult to discern a year horizon in 1988 in this core, and further investigation and possible reanalysis of the chemistry data for this year in this core is required before including this data in future analyses. McMorrow et al. (2001) suggest that snowfall events at LD have sufficient frequency to preserve monthly signals (i.e., linear interpolation errors are minimal). In addition, snowfall accumulation at LD is relatively uniform throughout the year, allowing seasonal LD records to be developed (Roberts et al., 2015). Previous studies have shown that the summer LD sea salt record preserves signals for MOCV, such as

the ENSO and the Interdecadal Pacific Oscillation (IPO; Vance et al. 2013; Vance et al. 2015), therefore we focus on the summer concentrations of sea salt from LD in this study.

To determine variability at the MBS site, and between the MBS and LD sites, the accumulation and sea salts in the individual MBS ice cores and the LD record were compared using Pearson's correlation coefficient. *P*-values for all analyses in this study



are based on effective degree of freedom ($N_{eff}$) calculated from a Lag-1 autocorrelation, according to Bretherton et al. (1999).
As the dating for the MBS record is still undergoing incremental improvement and may change in the future, analyses of the variability in accumulation and sea salts between the ice cores were repeated using a normalised Gaussian smooth (kernel = 3 yearly points, sigma = 0.6) to minimise the influence of timing inaccuracies within approximately one year.

## 2.5 Signals for modes of climate variability

To determine whether the MBS record preserves signals of regional or hemispheric climate variability, the degree of correlation
(lag-0) between the annual accumulation and sea salts against seasonal indices for the ENSO, SAM, and IOD was determined using Pearson's correlation coefficient. Summer sea salt concentrations from the LD record were also compared with seasonal MOCV indices, extending the analyses undertaken by Vance et al. (2013) from 2009 to 2016 to determine whether the ENSO signal in the LD record varies with a longer instrumental overlap period. Seasonal indices, rather than annual, were used as these MOCV have their own temporal cycles that do not match the calendar year. The SAM index was separated into seasons
(i.e., DJF, MAM, JJA, SON) as the SAM exhibits high frequency variability (weekly to seasonal). The ENSO indices were averaged over June-November as SSTA and convection anomalies related to the ENSO tend to emerge in early austral winter in the equatorial and SW Pacific and propagate to higher southern latitudes during austral spring and into summer (Fogt and Bromwich 2006; L'Heureux and Thompson 2006). In addition, we wished to align as closely as practicable to the seasonal cycle of sea salt concentrations at MBS, which is highest during austral autumn to spring (similar to LD) due to decreased
mid-latitude storminess in summer (Goodwin et al. 2004; see Appendix B). We averaged the IOD index over austral spring to align with the peak of IOD activity (Cai et al. 2013).

The ENSO indices used in this study include the Southern Oscillation Index (SOI; Parker 1983, available from http://www.bom.gov.au/climate/current/soi2.shtml), the Multivariate El Niño-Southern Oscillation Index (MEI; Wolter and
Timlin 2011, available from https://psl.noaa.gov/enso/mei/), and the Niño 3.4 and Niño 4 indices (Trenberth 1997, available from https://climatedataguide.ucar.edu/climate-data/nino-sst-indices-nino-12-3-34-4-oni-and-tni). The SAM index used in this study is the Marshall index (Marshall et al. 2003, available from https://legacy.bas.ac.uk/met/gjma/sam.html). The IOD index used in this study is the Dipole Mode Index (DMI; Saji et al. 1999, available from http://www.jamstec.go.jp/aplinfo/sintexf/iod/dipole_mode_index.html). The ENSO, SAM and IOD indices were detrended (as
were the corresponding accumulation and sea salts time series) to reduce the interference of any climate change signals and ensure any significance was due to inter-annual variability, rather than (for example) the pronounced shift toward the positive SAM phase during austral summer in recent decades (Marshall et al. 2003; Thompson and Wallace 2000).

## 2.6 Observational and reanalysis datasets

To investigate atmospheric processes leading to the preservation of any climate signals, KNMI Climate Explorer (see
https://climexp.knmi.nl/start.cgi) was used to create composite maps of SSTA, zonal winds and geopotential height (GPH)





during the upper and lower terciles of sea salt years in the MBS and LD records. These fields were detrended in order to focus on the inter-annual variability of the MOCVs. The SSTA were constructed using the HadISST observational dataset. The zonal wind and GPH anomalies were constructed using the ERA5 reanalysis dataset, which has an improved grid resolution (0.25º x 0.25º, ~27 km) compared to ERA-interim (0.75º x 0.75º, ~82 km; Tetzner et al. 2019). *P*-values used for hypothesis

significance testing were computed to indicate regions where anomalies are significant at the 90% level. Although the composite years are based on annual sea salts, the months displayed in the composite maps were chosen based on the months found to have significant correlations with the relevant MOCV.

To determine the reliability of the snowfall accumulation rates derived from the independent layer-counting, the accumulation

rates for the MBS ice cores were compared to the annual snowfall precipitation from ERA5. The weighted average precipitation from the four neighbouring geographic pixels were used to represent accumulation at the Main ice core site (see Appendix B). As high wind speeds can remove snow, we also sought to determine whether snowfall precipitation from ERA5 would be a useful comparison to ice core derived accumulation in the first case. Guided by the findings of Li and Pomeroy (1997) who determined a dry snow saltation wind speed threshold of 7.7 m s$^{-1}$, we used coincident ERA5 near-surface wind to investigate

the presence of a threshold for wind removal of fresh snow. We computed the correlation between annual total ice core accumulation and annual total ERA5 precipitation occurring during times when the wind speed was lower than a particular threshold. We performed this sensitivity test for a range of wind speeds from 0 to 30 m s$^{-1}$.

### 3.0 Results

### 3.1 Mount Brown South site features 1975-2016

The MBS record has a lower mean sample resolution and accumulation (15 samples per year, 0.296 ± 0.06 m yr$^{-1}$ IE, and sample sizes of 3 cm) compared to the LD record (24 samples per year, 0.747 ± 0.15 m yr$^{-1}$ IE, and sample sizes of 5 cm) over the period 1975-2016. The currently analysed sections of the Charlie ice core covers the period 1975-2016, the Alpha core covers 1979-2016, and the Main core covers 1975-2007 (extends back likely 1,000 years however, only the upper section has been dated at this stage and is considered here). The new short LD ice core (that extends the satellite era record) covers the

period 1989-2016, meaning that it replaces and extends the four short ice cores used in Vance at al. (2013). A peak in nssSO$_4^{2-}$ recorded in the MBS ice cores aligns with the reference horizon for the Pinatubo eruption which occurred in mid-1991. Plummer et al. (2012) dated the sulphate deposition event attributed to Pinatubo in the LD ice core as commencing in 1991.7 and ending around 1993.9 (Plummer et al. 2012). The MBS cores presented here show reference horizons attributed to the Pinatubo eruption in late 1991 (Alpha) and in 1992 (Charlie and Main; see Fig 2.). Prior to this volcanic signature are 4-5 years

where the annual horizons are harder discern. The period 1986-1990 contains isotopic and trace chemistry signatures that are at times ambiguous. This period is similarly difficult to date in multiple LD records, and is likely due to a period of extensive atmospheric blocking that occurred around this period in the south Tasman Sea that led to warm intrusions of moisture (Pook





and Gibson 1999). Atmospheric blocking and subsequent moisture intrusions can deliver significant snowfall to East Antarctica (Masson et al. 2004; Wille et al. 2019). The LD composite core has had extensive examination over multiple

overlapping records of the Pinatubo eruption, thus we assume the correct date for MBS reference horizons in coastal East Antarctica is likely to be late 1991-1993. There is the possibility of a 1 year dating error in the MBS composite records; alternatively, it could be that the main body of the sulphate deposition for MBS occurred later at the lower accumulation MBS site. Either way, further analyses once available (e.g. hydrogen peroxide and potentially volcanic dust) may help limit any dating errors from the interim suite of analytes available for layer-counting for this study.


The mean annual snowfall accumulation rates determined in the Alpha, Charlie, and Main ice cores are $0.298 \pm 0.07$, $0.295 \pm 0.08$, and $0.309 \pm 0.08$ m yr$^{-1}$ IE, respectively. The sensitivity test showed that the effect of wind-blown snow at MBS was negligible. The accumulation in the three MBS ice cores are significantly correlated in all cases, and these correlations are increased after smoothing (see Fig. 3). The MBS Alpha and MBS site average accumulation records are significantly correlated

with the LD accumulation record. The ERA5 estimate of annual snowfall accumulation rate for the MBS site is $0.302 \pm 0.05$ m yr$^{-1}$ IE. The degree of correlation between the MBS average record with ERA5 data is high and significant, especially when considering the independence of the two techniques (atmospheric reanalysis-derived site average versus independent layer counting). The ERA5 accumulation estimate suggests two consecutive years of low accumulation may have occurred in sequence (e.g., 1993, 1994), which is not demonstrated in the layer-counted data.


The mean annual sea salt concentrations (not log-transformed) in the Alpha, Charlie and Main ice cores and the MBS site average are $1.085 \pm 0.32$, $1.052 \pm 0.38$, $1.118 \pm 0.36$, and $1.090 \pm 0.31$ µEq l$^{-1}$, respectively, while the mean annual sea salt concentration in the LD record is $3.280 \pm 1.32$ µEq l$^{-1}$. The annual log-transformed sea salt concentrations from the MBS ice cores (site-averaged) and the LD record are not significantly correlated, although the annual sea salt concentrations in the MBS

record and the summer sea salt concentrations from the LD record are significantly (and negatively; see Fig. 4) correlated. These correlations between the three MBS ice cores increase after smoothing. Note the outlier (Charlie core, 1987) with data that requires re-analysis and further investigation prior to inclusion in the timeseries.

### 3.2 Signal for the El Niño Southern Oscillation

The MBS sea salt record is significantly correlated with the seasonal MEI, Niño 4, Niño 3.4, and the SOI indices (see Table

1). The LD summer sea salt record is also significantly correlated with the seasonal Niño 4, Niño 3.4, and SOI indices, and is marginally significant (*p*-value 0.051) for the MEI. The MBS-ENSO correlation (positive correlation with SST indices) is higher and more significant compared to the LD-ENSO correlation (negative correlation with SST indices). For example, the highest *r*-value between the annual MBS sea salt record and the seasonal ENSO index is $r = 0.533$, $p = 0.001$ (MEI), whereas the highest *r*-value between the summer LD sea salt record and the seasonal ENSO index is $r = 0.393$, $p = 0.011$ (SOI). The

accumulation records from the MBS and LD records are not significantly correlated with any of the MOCV indices. Neither





the MBS nor the LD accumulation records are significantly correlated with the DMI representing the IOD, despite the location of MBS in the southern Indian Ocean sector of Antarctica.

The MBS sea salt record does not capture all ENSO events however, it does capture some of the larger La Niña events over
the period 1975-2016, for example, the 1974/75 and 2010/11 La Niña events are associated with troughs in sea salt concentrations (see Fig. 5). Similarly, the 1982/1983, 1987/1988 and 2009/2010 El Niño events have corresponding peaks in the MBS sea salt record, however the extreme 1997/1998 event is less clear, with a higher sea salt peak one year prior. The scatterplot suggests extreme anomalies (i.e., ENSO events) are reasonably well represented by the MBS sea salts, in contrast to LD sea salts in Vance et al. (2013), which is skewed toward a representation of La Niña rather than El Niño events.

Composite maps of SSTA in June-November based on high (low) MBS sea salt years indicate significant warming (cooling) of the central and eastern equatorial Pacific (see Fig. 6b and 6d). Composite maps in December-March based on the LD sea salts indicate an opposite trend in SSTA (i.e., warming (cooling) in the central and eastern equatorial Pacific during the low (high) LD summer sea salt years) however, these anomalies are not significant (see Fig. 6a and 6c).

The composite map of GPH anomalies at 500 hPa (Z500) in June-November based on high MBS sea salt years corresponds with a significant weakening of the Amundsen Sea Low (ASL; see Fig. 7b), while low sea salt concentration years correspond with a significant strengthening of the ASL (note this anomaly is shifted toward the west; see Fig. 7e). Composite maps in November-February based on the LD record indicate a significant opposite pattern in GPH anomalies (i.e., strengthening
(weakening) of the ASL during the high (low) LD summer sea salt concentration years; see Fig. 7a and 7d). Composite maps based on the Niño 3.4 index show similar GPH anomalies to those based on the MBS record, although they are stronger and more widespread.

Fig. 8 shows composite anomaly maps of the 10 m wind in June-August (austral winter) and September-November (spring)
based on high and low sea salt years, along with maps of the mean 10 m wind. The prevailing wind direction in the MBS region is from the south-east in these seasons (see Fig. 8a and 8b). A feature of the winter pattern for the low and high sea salt years (Fig. 8e and 8c, respectively) is the strong circulation changes north of the MBS region, at around 60° S. In winter during low sea salt years (see Fig. 8e), there is a weakening of the prevailing westerly flow associated with general circulation around an anomalous cyclonic feature to the north of MBS, and an anomalous strengthening of the prevailing easterlies closer to the
MBS site. In winter during high sea salt years (see Fig. 8c), there is a strengthening of the prevailing westerly flow around an anomalous anti-cyclonic feature, and an anomalous weakening of the polar easterlies. This anomaly is centred over the ice edge in winter. We suggest the ability of moisture-carrying sea salt aerosols to become advected over the MBS site is weaker during low sea salt years, and stronger during high sea salt years. In the spring, the off-shore and on-shore patterns are weaker (see Fig. 8d and 8f) and their potential influence on frontal passages is less clear.



### 3.3 The Southern Annular Mode and Indian Ocean Dipole

Table 1 shows that the MBS sea salt record is not significantly correlated with the DMI (IOD) or Marshall (SAM) indices. The LD summer sea salt record is not significantly correlated with the Marshall index during December-March. These months are based on those from Vance et al. (2013). Despite the lack of a SAM signal in the annual MBS sea salt record, composite maps show SAM-like patterns in 500 hPa zonal wind anomalies during austral summer. Composites of zonal wind anomalies at this height were chosen to reduce the influence of katabatic winds and topography. Composite maps in December-February based on low MBS sea salt years indicate a significant enhancement of zonal wind anomalies in the high-latitudes and coincident reduction in the mid-latitudes (see Fig. 9e). The converse is true for composite maps based on the high MBS sea salt years with a significant reduction of zonal wind anomalies in the high-latitudes and enhancement in the mid-latitudes, although the reduction in high-latitude zonal wind anomalies is not symmetrical across the southern hemisphere (see Fig. 9b). Composite maps in November-February based on the LD summer sea salt years indicate an opposite and significant pattern in zonal wind anomalies (i.e., enhancement (reduction) of zonal wind anomalies in the high- (mid-) latitudes during low LD sea salt years (see Fig. 9d), and vice versa for high sea salt years (see Fig. 9a). Composite maps based on the SAM Marshall index show opposite zonal wind anomalies to those based on the MBS record, although with a more zonally symmetric pattern in the upper tercile years.

### 4.0 Discussion

### 4.1 Mount Brown South ice core statistics 1975-2016

The currently dated recent portion of the MBS record fulfils the requirements stated by Vance et al. (2016) of having an annual resolution (i.e., accumulation rate > 0.25 m yr$^{-1}$ IE), a strong teleconnection with the low- and mid-latitudes, and based on the initial surface accumulation rate analysis presented here, will likely date back 1,000 years at 300 m depth. The mean accumulation rate at MBS (0.297 ± 0.06 m yr$^{-1}$ IE) is identical to the findings of Foster et al. (2006; 0.297 m yr$^{-1}$ IE over the period 1984-1999), higher than the findings of Smith et al. (2002; 0.255 m yr$^{-1}$ IE, maximum of 10 years over the period 1976-1998), and higher than estimated in Vance et al. (2016; 0.200 - 0.250 m yr$^{-1}$ IE, over the period 1989-2009) suggesting that the airborne radar estimates underestimate net surface mass balance in this region. There is some variability in individual accumulation years between the MBS ice cores but not enough to suggest that the relatively large sastrugi and drifts (as high as 0.5 m, e.g., more than the annual accumulation rate derived for MBS) present during the drilling process are a common feature at the site on inter-annual timescales (see Appendix C). The mean accumulation rate in the new LD record (0.747 ± 0.15 m yr$^{-1}$ IE) is higher than presented in Morgan et al. (1997; 0.678 m yr$^{-1}$), Roberts et al. (2015; 0.688 m yr$^{-1}$), and van Ommen and Morgan (2010; 0.688 m yr$^{-1}$) however, these studies focus on longer time periods (~2,000 years). The higher accumulation rate in recent decades compared to the longer-term average supports the findings of van Ommen and Morgan (2010) and Medley and Thomas (2018). The lower accumulation rate in the MBS record compared to the LD record means



that MBS may incur more dating error if specific seasonal features such as water stable isotopes are not preserved. Although, the high *r*-values between the ERA5 precipitation data and the snowfall accumulation from independently dated MBS ice cores suggests that the dating is reliable, at least during the period examined here (1975-2016). The interim dating of the MBS ice cores via independent layer counting means that they are not corrected with ERA5 data, rather ERA5 is used to determine how and why the independent layer counting may be wrong. This is an important first step to understand the errors of layer counting in the MBS ice cores. Further analyses (e.g. hydrogen peroxide, dust and tephra) will help to discern correct attribution of annual layers (hydrogen peroxide), volcanic deposition (dust) and tephra identification (tephra).

The MBS record preserves signals for the ENSO, indicating that the record does contain signals for past climate variability beyond the local climate conditions. The MBS record differs from the LD record in annual sea salt concentrations, and provides another independent climate record for the data sparse Indian Ocean sector of Antarctica. The climatology of the sea salt concentrations in the MBS record and updated LD record (see Appendix D) are comparable with the LD record presented in Curran et al. (2003), and potentially result from the increase in mid-latitude storminess during winter (Goodwin et al. 2004). However, the lower sample resolution in the MBS record means that the climatology of sea salts should be interpreted with caution until the periodicity of snowfall events at MBS is determined and a longer record is developed. Interestingly, the annual MBS record and summer LD record are negatively correlated. This may be due to the fact that annual sea salt deposition at LD is heavily dominated by winter coastal storminess that may drown out regional signals, whereas the summer sea salt deposition is less influenced by the local climate (i.e., storminess decreases), and therefore regional signals may emerge. The LD and MBS snowfall accumulation rate records are also significantly correlated. The degree of correlation is not large (e.g. explaining around 15% of the variance at most), however it suggests both records share aspects of a similar atmospheric 'catchment' of the southern Indian Ocean. This again suggests that (similar to LD) MBS preserves regional climate signals, rather than just local climatology, and understanding how the preservation of climate signals occurs will benefit from concurrent synoptic typing and moisture transport studies for the southern Indian Ocean (e.g. Udy et al., in review; Gorodetskaya et al. 2014).

### 4.2 Preservation of the El Niño Southern Oscillation signal in the Mount Brown South record

The MBS record contains a stronger signal for the ENSO compared to the LD record, including the LD record presented in Vance et al. (2013; $r = 0.336$, $p < 0.000$) and the updated record presented here. The change in the strength of the ENSO signal in the LD record over time may be due to decadal climate variability caused by the IPO, as suggested in Vance et al. (2013). The IPO is a low-frequency climate oscillation (20-30 years) characterised by warm (cool) SSTA in the equatorial Pacific during the positive (negative) IPO phase (Power et al. 1999). It is debated whether the IPO is a low frequency ENSO (e.g., Westra et al. 2015) or an independent climate mode (e.g., Newman et al. 2016). The interaction between the ENSO and IPO are not fully understood, although it is suggested that the IPO influences the strength and locality of the teleconnection between the equatorial Pacific and the southern high-latitudes (Vance et al. 2013; 2015; 2016). Although the timescale of this study is



too short to investigate decadal climate variability, it is possible that the IPO also influences the ENSO signal in the MBS
record. A few studies have detected a signature of the IPO in the southern Indian Ocean (Crueger et al. 2009; Vance et al.,
2015; 2016). Further studies investigating this may aid in correctly interpreting the MBS record.

Although fully understanding the mechanisms behind the transport of the ENSO signal to MBS requires further analysis,
preliminary analyses of oceanic and atmospheric processes suggests that the ENSO signal is robust. SSTA based on the MBS
record reflect patterns associated with El Niño and La Niña events, and are supported by the correlation results (i.e., low sea
salt concentration years occur during La Niña events and are associated with low SSTA in the equatorial Pacific, and vice
versa). GPH anomalies based on the MBS record reflect patterns associated with El Niño and La Niña events, specifically
changes in the strength of the ASL. This pattern is evident in our results, further supporting the reliability of the ENSO signal
in the MBS record. Composite maps based on the LD record produce opposite patterns in SSTA and in GPH anomalies
compared to the MBS record, which is expected as these records are inversely correlated (note summer LD versus annual MBS
sea salt records). The GPH anomalies based on the LD summer sea salt record are skewed toward low sea salt years (i.e., El
Niño-like years). This supports the findings of Vance et al. (2013) that suggest the LD record has a weaker response to La
Niña compared to El Niño events.


It is difficult to directly compare the ENSO signals at MBS and LD because the MBS signal is based on annual sea salt
concentrations, which are heavily influenced by winter storminess, whereas the LD signal is based on summer sea salt
concentrations. The dating of annual sea salts is easier to constrain compared to summer sea salts (due to the assumption of
uniform accumulation to define seasonal boundaries). Concurrent work examining the synoptic control of moisture transport
to East Antarctic ice core sites will help to define seasonal records from MBS (Udy et al., in review), while future work is
planned to investigate the influence of atmospheric rivers on moisture (and sea salt) transport to MBS and LD (e.g. following
Gorodetskaya et al. 2014; Wille et al., 2019). Udy et al. (in review), found coherence between ENSO events and synoptic
variability in the southern Indian Ocean during austral spring and summer. Specifically, this coherence related to synoptic
types that had clear meridional structures and were related to blocking in the southern Indian Ocean in contrast to synoptic
types with a more zonal structure, hinting at the dynamic cause of the ENSO signals at MBS and LD.

It was noted during the drilling process that the sastrugi at MBS were aligned to approximately 80º/260º, suggesting the
prevailing winds during the 2017/18 austral summer were likely east-north-easterlies (see Appendix E), although as mentioned
before this summer may have been anomalous. If however, the prevailing winds at MBS are east-north-easterlies, it is possible
that the maritime signal is less influenced by katabatic winds, and therefore the MBS record may contain a stronger maritime
signal compared to LD, at least in terms of moisture delivery to MBS. The prevalence of the maritime signal is also observed
in the precipitation impact from atmospheric rivers, which provide a subtropical link to the Antarctic continent. Atmospheric
rivers have been linked with extreme precipitation events in East Antarctica dramatically altering the regional SMB and raising





questions about their influence on paleoclimate records (Gorodetskaya et al. 2014). Such extreme precipitation events also

have a dominant role in Antarctic snowfall variability (Turner et al. 2019). Atmospheric river landfalls across Antarctica are typically associated with a blocking ridge upstream of the landfall location and intense meridional moisture fluxes (Wille et al. 2019; Gorodetskaya et al. 2020). Future studies focusing on back trajectories of wind direction/speed, atmospheric river events, and the prevalence of atmospheric blocking in the MBS region and comparing these with previous back trajectory studies (e.g., Scarchilli et al. (2011)) and synoptic typing studies (e.g. Udy et al., (in review)) are recommended to improve the

quantitative interpretation of the record.

Overall, we suggest that the main contrast between the troughs and peaks in the MBS annual sea salt record relate more strongly to the circulation anomalies that develop in austral winter than spring, and we hypothesise that this relates to the ability of frontal systems to transport regionally sourced sea salt aerosol via increases in surface wind speeds over the ocean (or frost

flowers) to the north of MBS to the site. Whether frost flowers contribute to the sea salt aerosols that are deposited at MBS is yet to be determined. We also suggest that the particular off-shore circulation features shown in Fig. 8c and 8e are influenced more by a teleconnection related to the ENSO than from other MOCVs.

### 4.3 Lack of preservation of a Southern Annular Mode signal in the Mount Brown South record

Although the annual MBS record does not preserve a strong SAM signal, it is possible that better dating in the future may

allow for the investigation of a seasonal SAM signal using austral summer or other seasonal sea salts. Composite maps based on the MBS record reflect SAM-like patterns in zonal wind anomalies (i.e., the contraction and the expansion of the westerly wind belt around the Antarctic continent; Marshall et al. 2003) in austral summer, which suggests that the MBS record may contain a SAM signal during summer. These patterns in zonal wind anomalies are not dependent on the trend in summer SAM since 1958 (Marshall 2003; Thompson et al. 2000), as both the Marshall index and corresponding sea salts were detrended

here. The new updated LD record does not preserve a SAM signal using the Marshall index months presented in Vance et al. (2013), although the study here only compares ice core data to indices since 1975, in contrast to 1958-2009 in Vance et al., (2013). This may be due to differences in timing noise between individual ice cores that make up the record or may result from a decadal signal in how the SAM signal is preserved (e.g., stronger relationship earlier in the index). Composite maps based on the LD record show opposite patterns in zonal wind anomalies compared to the MBS record, and these are skewed towards

low sea salt years (negative SAM-like years). This is expected as Marshall et al. (2017) suggest that the LD contains a skewed SAM signal, with increased precipitation in the LD region associated with the negative SAM phase.

Although the SAM-like patterns in zonal wind anomalies based on the MBS and LD record are approximately symmetrical, there are multiple studies debating the symmetry of SAM. Fogt et al. (2012a) suggest asymmetries in the SAM are related to

the ENSO (in austral spring and summer) and the Zonal Wave 3 (in winter), and associated with shifts in the entrance and exit regions of the Antarctic jet stream in the South Pacific. Ding et al. (2012) suggest that the SAM signature in the Indian Ocean





and South Pacific sectors of Antarctica differ. The authors describe the form of SAM in the Indian Ocean as an elongated meridional dipole due to strong internal dynamics, and the form of SAM in the South Pacific as a longitudinal band due to forcing from equatorial Pacific SSTA. Asymmetries in the SAM may be related to increased or decreased prevalence of

blocking events, which may influence the accumulation at MBS and LD as blocking events control extreme moisture intrusion in the form of atmospheric rivers (e.g., Wille et al. 2019). Scott et al. (2019) suggest that El Niño and negative SAM conditions favoured blocking events in the Amundsen Sea, both leading to warm surface air anomalies in West Antarctica. The influence of warm, moist intrusions on East Antarctica is less understood, although further atmospheric analyses in the southern Indian Ocean/MBS region may aid in explaining the ENSO teleconnection, despite the remoteness of MBS from the Pacific.

### 500 4.4 Interactions between the El Niño Southern Oscillation and Southern Annular Mode

Multiple studies have documented interactions between the ENSO and SAM (e.g., Cai et al. 2011; Clem and Fogt 2013; Dätwyler et al. 2020; Fogt et al., 2011; Gong et al. 2010; Kim et al. 2017; L'Heureux and Thompson 2006; Lim et al. 2013; Yu et al. 2015). A recent study by Dätwyler et al. (2020) looking at millennial timescale climate variability showed that the ENSO and SAM are significantly correlated during austral summer ($r = -0.3$). This negative correlation arises when the ENSO

and SAM are in phase (i.e., when El Niño-negative SAM or La Niña-positive SAM co-occur). When El Niño-negative SAM (La Niña-positive SAM) events coincide, storm tracks shift equatorward (poleward) resulting in the weakening (strengthening) of circumpolar westerly winds (Dätwyler et al. 2020). Fogt et al. (2011) suggest that the SAM modulates the ENSO teleconnection with the southern high-latitudes. When the ENSO and SAM are in phase, transient eddies in the mid-latitudes are supportive, which reinforces the ENSO teleconnection. In contrast, when the ENSO and SAM are out of phase (i.e., when

El Niño-positive SAM or La Niña-negative SAM co-occur), these transient eddies oppose each other, weakening the ENSO teleconnection. Furthermore, L'Heureux and Thompson (2006) show that the ENSO accounts for 25% of the temporal variability in the SAM during austral summer.

Studies on the interaction between the ENSO and SAM tend to focus on the interaction between these MOCVs in the South

Pacific at the southern end of the PSA pressure pattern. In the Indian Ocean sector of the Southern Ocean, adjacent to where MBS is located, the ENSO and SAM interaction is less well understood. Further studies investigating whether there is a significant SAM signal preserved in the longer, seasonal MBS chemistry and snowfall accumulation records, and the interaction of these two signals in the southern Indian Ocean are important for correctly interpreting the complete MBS record.

### 5.0 Conclusion

The upper portion of the MBS ice core preserves a statistically significant signal for the ENSO (in austral winter and spring) in the annual sea salt concentrations. The ENSO signal in the MBS record is stronger than that found in the LD record, including both the record presented in Vance et al. (2013) and the updated record presented here. Investigation into the environmental conditions leading to the preservation of the ENSO signal in the MBS record indicates ENSO-like patterns in SSTA in the



equatorial Pacific, and changes in the strength of the Amundsen Sea Low. Surface wind anomalies over the ocean to the north

of MBS are likely to control increases or decreases in the amount of sea salt preserved in the annual layers of the MBS record, and hence may be the link between local/regional wind anomalies and production of sea salt aerosol, and the signal preservation of the larger MOCVs (i.e., the ENSO, potentially the SAM in future studies). Here, we focus on the period 1975-2016 however, the MBS record is estimated to span the last millennium. The extended MBS record will be used to generate a millennial-timescale ENSO proxy with established links to regional variability in the southern Indian Ocean that will improve the

reliability of past climate reconstructions, and hence future climate projections. Future work will investigate these potential signals of regional climate variability and whether a SAM or IOD signal is preserved in the seasonal trace ion chemistry or longer snowfall accumulation records from MBS, as these MOCVs affect temperature and hydroclimate across the Southern Hemisphere continents.




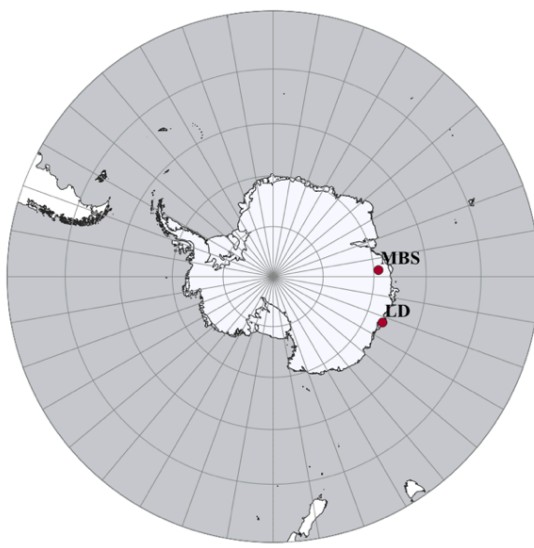

**Figure 1. Map of the Mount Brown South (MBS) and Law Dome (LD, Dome Summit South site) sites in East Antarctica.**





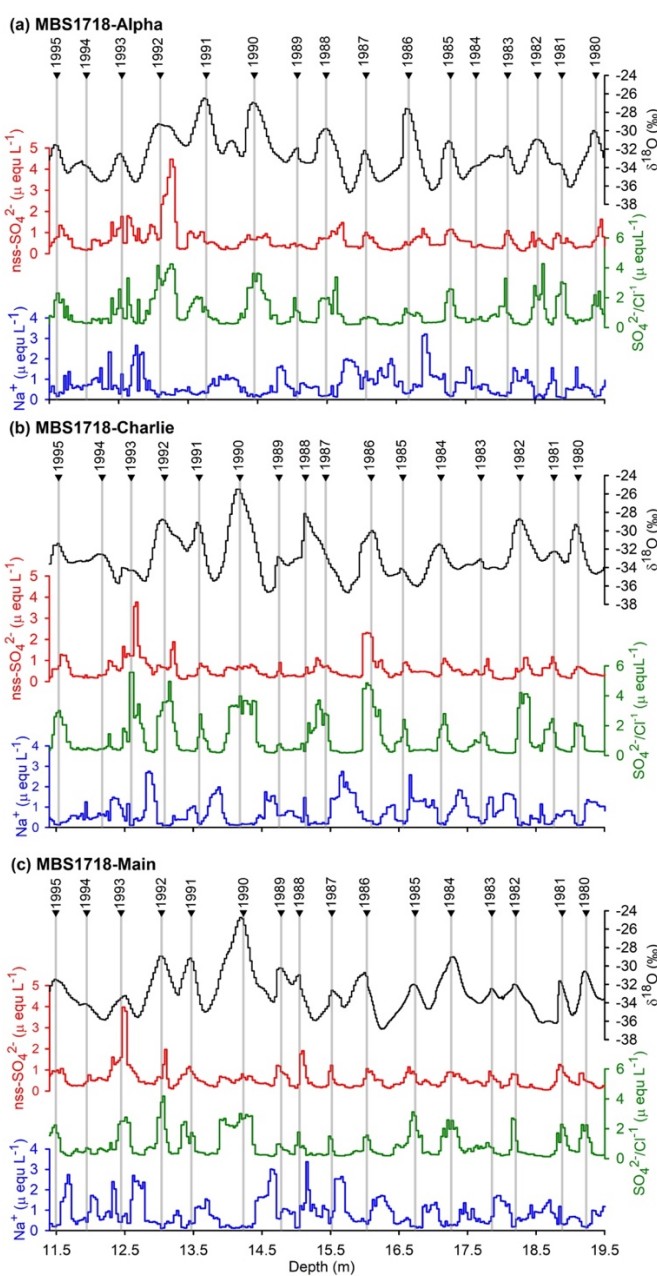


**Figure 2. An 8 m long section from the Mount Brown South (MBS1718-Alpha (a), MBS1718-Charlie (b) and MBS1718-Main (c)) ice cores. These sections cover the period 1980-1995 (which contains some ambiguous annual horizons, see section 3.1), with annual layers (year boundaries) shown as triangles with vertical grey lines. Annual horizons are identified from summer-peaking glaciochemical species (i.e., oxygen isotopes ($\delta^{18}$O), non-sea salt sulphate (nssSO$_4^{2-}$)**

**and the ratio between sulphate and chloride (SO$_4^{2-}$/Cl$^-$)). Winter-peaking sea salts (Na$^+$) are used as a confirmatory species. Panels (a) and (b) are on the same depth scale as shown in (c).**



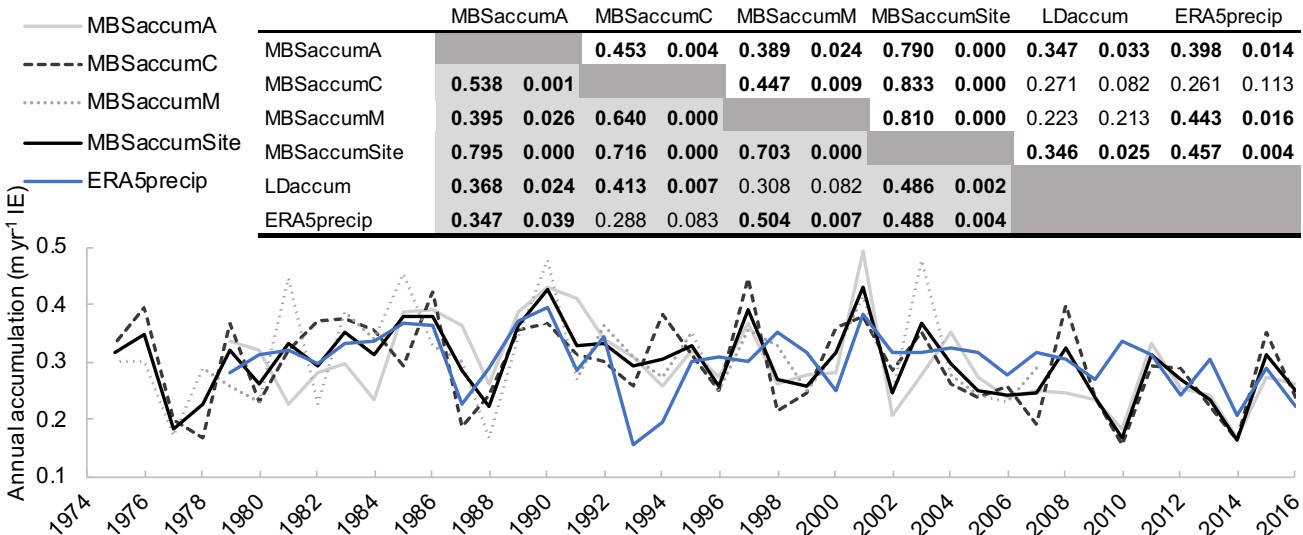

**Figure 3. Timeseries of annual snowfall accumulation for the Mount Brown South site average (MBSaccumSite), the Alpha (MBSaccumA), Charlie (MBSaccumC) and Main (MBSaccumM) ice cores, and the ERA5 precipitation data**
**from the MBS Main site (ERA5precip). Embedded table shows Pearson's correlation coefficient (*r*-left column, *p*-right column) for the average annual snowfall accumulation for MBSaccumSite, MBSaccumA, MBSaccumC and MBSaccumM, and the annual snowfall accumulation for the Law Dome (Dome Summit South site) record (LDaccum). Correlations using Gaussian smoothed (kernel = 3 yearly points, sigma = 0.6) accumulation time series are shaded and presented in the lower-left of the table. Correlations significant at 95% are bold, and the dates range between 1975-**
**2016.**





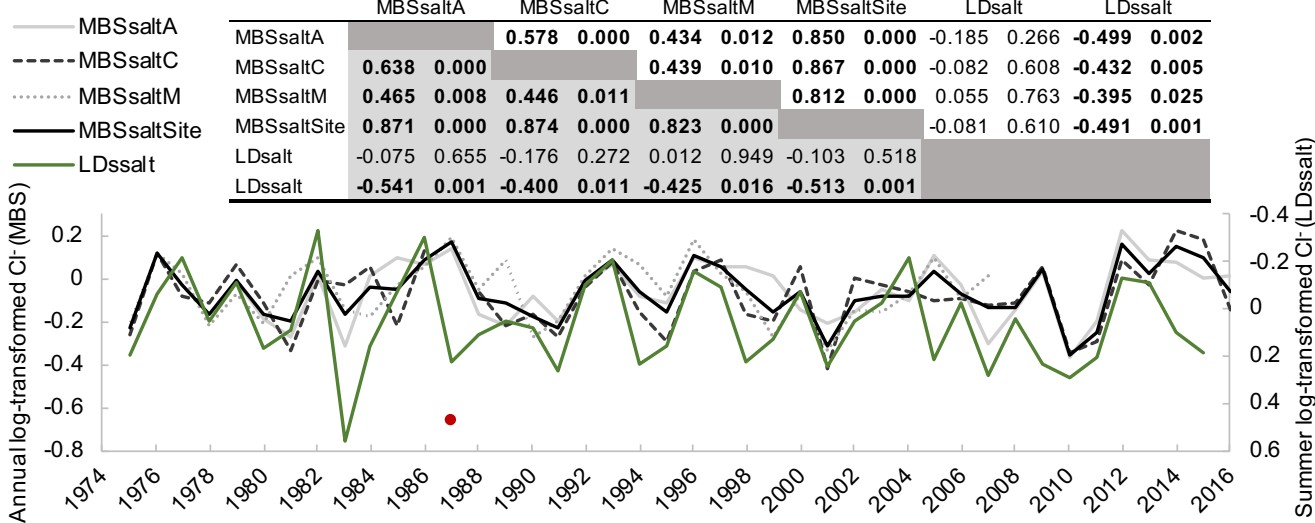

**Figure 4. Timeseries of log-transformed annual sea salt concentrations for the Mount Brown South site average**

**(MBSsalt Site), the Alpha (MBSsaltA), Charlie (MBSsaltC) and Main (MBSsaltM) ice cores, including the Charlie 1987 (red dot) outlier, and the log-transformed summer sea salt concentrations for the Law Dome (Dome Summit South site; LDssalt) record (on inverted secondary axis). Embedded table shows Pearson's correlation coefficient (*r*-left column, *p*-right column) for the log-transformed sea salt concentrations for MBSsaltSite, MBSsaltA, MBSsaltC, MBSsaltM, LDssalt, and the log-transformed annual sea salt concentrations for the Law Dome (LDsalt) record. Correlations for**

**normalised Gaussian smoothed (kernel = 3 yearly points, sigma = 0.6) sea salt concentrations are shaded and presented in the lower-left of the table. Correlations significant at 95% are bold, and the dates range between 1975-2016.**





**Table 1. Pearson's correlation coefficient for the detrended accumulation (MBSaccumSite) and detrended annual log-**
**transformed sea salt concentrations for the Mount Brown South site average (MBSsaltSite), and the detrended summer**
**log-transformed sea salt concentration (LDssalt) for the Law Dome (Dome Summit South site) record against detrended**
**seasonal ENSO, SAM, and IOD indices. Correlations significant at 95% are bold, and the dates range between 1975-**
**2016.**

|  |  | Seasonal indices | Range | R-value | P-value |
|---|---|---|---|---|---|
| MBSsaltSite | **ENSO (MEI)** | **JJASON** | **1979 - 2016** | **0.533** | **0.001** |
|  | **ENSO (Niño 3.4)** | **JJASON** | **1975 - 2016** | **0.521** | **0.000** |
|  | **ENSO (Niño 4)** | **JJASON** | **1976 - 2016** | **0.457** | **0.002** |
|  | **ENSO (SOI)** | **JJASON** | **1977 - 2016** | **-0.496** | **0.001** |
|  | SAM (Marshall) | DJFcausal | 1978 - 2016 | 0.107 | 0.498 |
|  | SAM (Marshall) | MAM | 1979 - 2016 | 0.274 | 0.079 |
|  | SAM (Marshall) | JJA | 1980 - 2016 | 0.175 | 0.266 |
|  | SAM (Marshall) | SON | 1981 - 2016 | -0.168 | 0.287 |
|  | SAM (Marshall) | DJFacausal | 1982 - 2016 | -0.203 | 0.196 |
|  | IOD (DMI) | SON | 1983 - 2016 | 0.231 | 0.140 |
| LDssalt | ENSO (MEI) | SO | 1979 - 2015 | -0.323 | 0.051 |
|  | **ENSO (Niño 3.4)** | **SO** | **1975 - 2015** | **-0.387** | **0.018** |
|  | **ENSO (Niño 4)** | **SO** | **1975 - 2015** | **-0.333** | **0.033** |
|  | **ENSO (SOI)** | **NDJF** | **1976 - 2015** | **0.393** | **0.011** |
|  | SAM (Marshall) | DJFM | 1977 - 2015 | 0.122 | 0.447 |
|  | IOD (DMI) | SON | 1978 - 2015 | -0.202 | 0.206 |
| MBSaccumSite | ENSO (MEI) | JJASON | 1979 - 2016 | 0.226 | 0.173 |
|  | ENSO (Niño 3.4) | JJASON | 1975 - 2016 | 0.213 | 0.175 |
|  | ENSO (Niño 4) | JJASON | 1976 - 2016 | 0.147 | 0.353 |
|  | ENSO (SOI) | JJASON | 1977 - 2016 | -0.104 | 0.511 |
|  | SAM (Marshall) | DJFcausal | 1978 - 2016 | 0.150 | 0.342 |
|  | SAM (Marshall) | MAM | 1979 - 2016 | 0.065 | 0.684 |
|  | SAM (Marshall) | JJA | 1980 - 2016 | 0.043 | 0.787 |
|  | SAM (Marshall) | SON | 1981 - 2016 | 0.159 | 0.313 |
|  | SAM (Marshall) | DJFacausal | 1982 - 2016 | -0.145 | 0.361 |
|  | IOD (DMI) | SON | 1983 - 2016 | 0.148 | 0.348 |






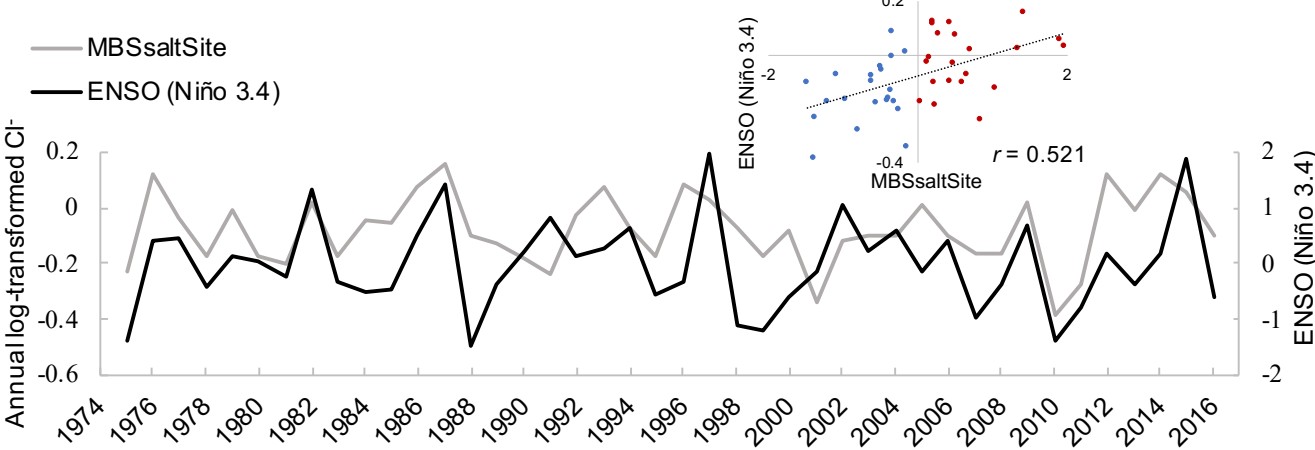

**Figure 5. Timeseries of the annual detrended, log-transformed sea salt concentrations for the Mount Brown South site average (MBSsaltSite) and the detrended (June-November) El Niño Southern Oscillation region SSTA (ENSO (Niño 3.4)). Embedded scatterplot shows the relationship between the two time series differentiated by positive (red dot, El Niño-like) and negative (blue dot, La Niña-like) Niño 3.4 region SSTA based on the period 1975-2016.**






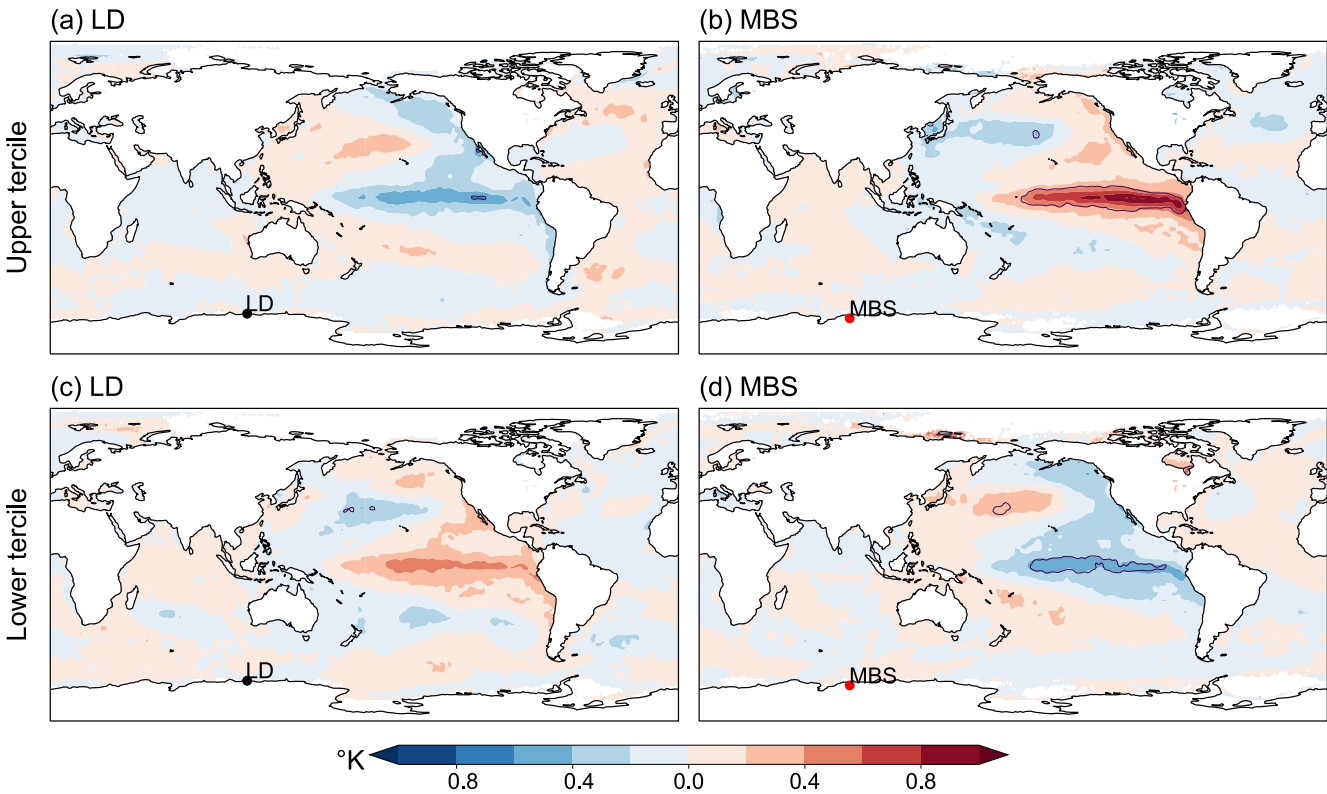

**Figure 6.** SSTA composite maps during upper (a) and lower (c) tercile (November-February) of the detrended, log-transformed summer sea salt concentrations from the Law Dome (LD, Dome Summit South site) record over 1975-2015, and upper (b) and (d) lower tercile (June-November) detrended, log-transformed sea salt concentrations from the annual Mount Brown South (MBS) site average over 1975-2016. The red dot indicates MBS and the black dot indicates LD. Significant anomalies are within the upper and lower purple contour lines, where $p < 0.1$.







**Figure 7. GPH composite anomaly maps (m at Z500) during upper (a) and lower (d) tercile (November-February) of the detrended, log-transformed summer sea salt concentrations from the Law Dome (LD, Dome Summit South site) record over 1979-2015. (b, e) same as (a, d) except using the detrended, log-transformed annual sea salt concentrations from the Mount Brown South (MBS) site average (June-November) over the period 1979-2016. (c, f) same as (b, e) except using the detrended Niño 3.4 index (June-November). The red dot indicates MBS and the black dot indicates LD. Significant anomalies are within the upper and lower purple contour lines, where *p* < 0.1.**





**Figure 8. Near-surface (10 m) wind vector composite maps of the mean state during austral winter (a, June-August) and spring (b, September-November), and anomaly composite maps during upper (c) and lower (e) tercile in June-August of the detrended, log-transformed annual sea salt concentrations from the Mount Brown South (MBS) site average, over the period 1979-2016. (d, f) same as (c, e) except in September-November. The red dot indicates the MBS site location. Latitudinal lines are in increments of 5º and longitudinal lines are in increments of 15º.**





**Figure 9. Zonal wind anomaly composite maps (m s⁻¹ at 500 hPa) during the upper (a) and lower (d) tercile (November-February) of the detrended, log-transformed summer sea salt concentrations from the Law Dome (LD, Dome Summit South site) record over the period 1979-2015. (b, e) same as (a, d) except using the detrended, log-transformed annual sea salt concentrations from the Mount Brown South site average (MBS, December-February) over the period 1979-2016. (c, f) same as (b, e) except using the detrended SAM Marshall index (December-February). The red dot indicates MBS and the black dot indicates LD. Significant anomalies are within the upper and lower purple contour lines, where *p* < 0.1.**





**Appendix**

**Appendix A. Pearson's correlation coefficient for the detrended, annual log-transformed sea salt concentrations (sodium) for the Mount Brown South site average (MBSsaltSite) against detrended seasonal ENSO, SAM, and IOD indices. Correlations significant at 95% are bold, and the dates range between 1975-2016. Where sodium concentrations were below detectable levels for ion chromatography, chloride concentrations were used with adjusted sea water ratios (6 samples in the LD sodium record were below detectable levels).**

|  | Seasonal indices | Range | R-value | P-value |
|---|---|---|---|---|
| **ENSO (MEI)** | **JJASON** | **1979 - 2016** | **0.503** | **0.001** |
| **ENSO (Niño 3.4)** | **JJASON** | **1975 - 2016** | **0.486** | **0.001** |
| **ENSO (Niño 4)** | **JJASON** | **1976 - 2016** | **0.408** | **0.007** |
| **ENSO (SOI)** | **JJASON** | **1977 - 2016** | **-0.454** | **0.002** |
| SAM (Marshall) | DJFcausal | 1978 - 2016 | 0.147 | 0.354 |
| SAM (Marshall) | MAM | 1979 - 2016 | 0.304 | 0.050 |
| SAM (Marshall) | JJA | 1980 - 2016 | 0.199 | 0.206 |
| SAM (Marshall) | SON | 1981 - 2016 | -0.139 | 0.378 |
| SAM (Marshall) | DJFacausal | 1982 - 2016 | -0.284 | 0.068 |
| IOD (DMI) | SON | 1983 - 2016 | 0.196 | 0.213 |

(MBSsaltSite)




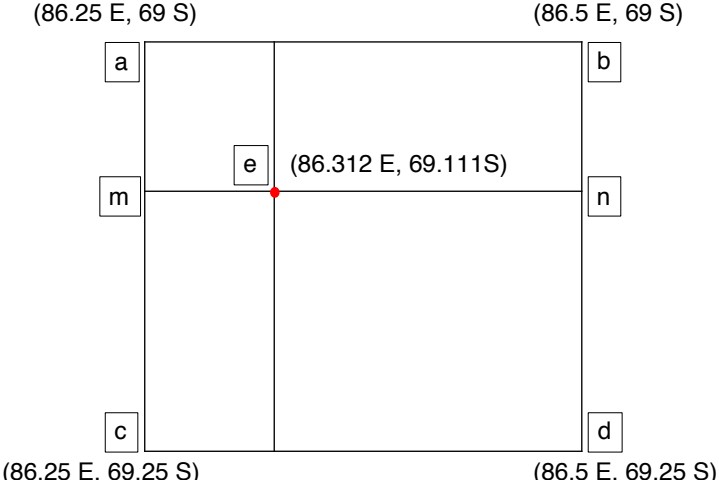

**Appendix B. The location of the MBS Main ice core site (e), and the four nearby ERA5 pixels (a, b, c, d). The weighted average of the a and c (m) and b and d (n) pixels were used to calculate the weighted average for the MBS Main site.**






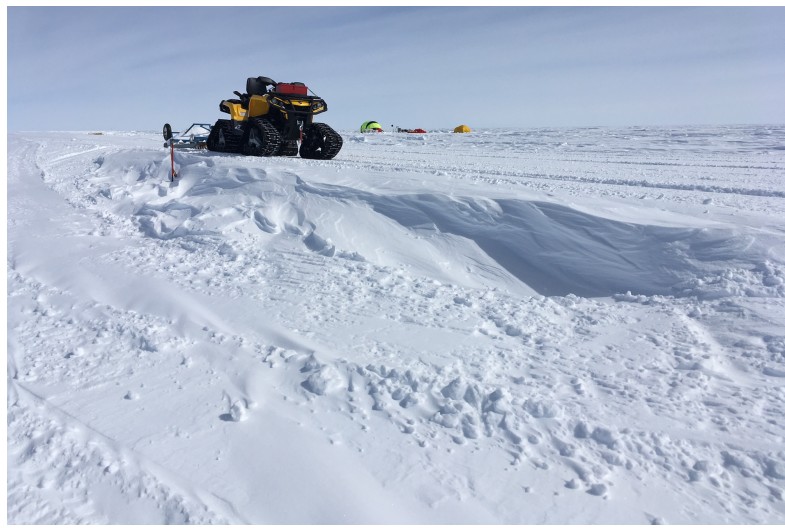

**Appendix C. Sastrugi (as high as 0.5 m) at Mount Brown South in East Antarctica during austral summer in 2017/18. The height of the largest sastrugi is comparable with the annual mean snowfall accumulation rate over the period 1975 – 2016.** *Photo credit: Tessa Vance.*





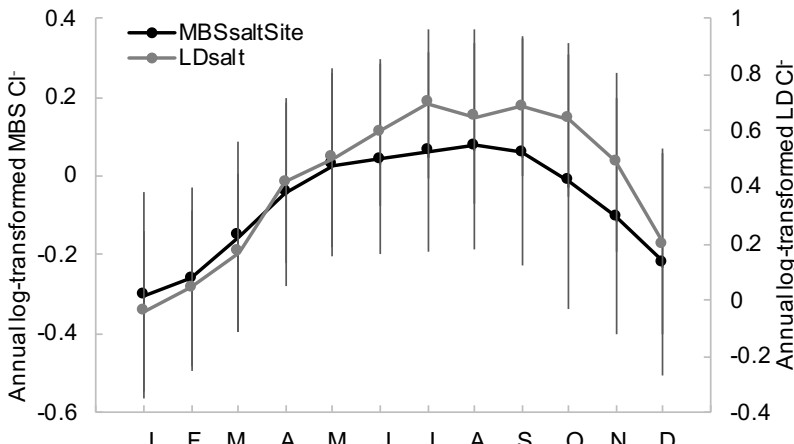

**Appendix D. Climatology of the annual log-transformed sea salt concentrations for the Mount Brown South site**
**average (MBSsaltSite) and Law Dome (Dome Summit South site; LDsalt) over the period 1975-2015/6. Error bar**
**vertical line plots give 1σ range on the composite monthly means.**



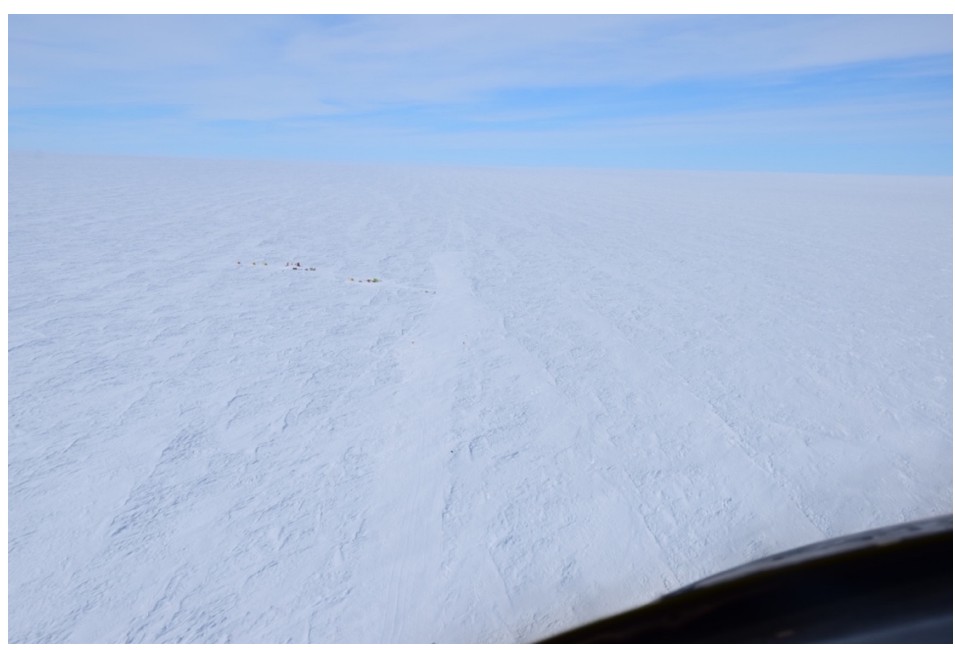

**Appendix E. Sastrugi (as high as 0.5 m) running in the west/east direction of approximately 80 degrees at Mount Brown South in East Antarctica during austral summer 2017/18 can be seen, particularly to the right of the image (a groomed skiway for aircraft at a slightly deeper angle of approximately 110 degrees can be seen in the centre of the image). This indicates that the prevailing winds during the summer 2017/18 season were east-north-easterlies.** *Photo Credit: Doug Westersund/Kenn Borek Air.*




**Data Availability**

Data uploaded to the Australian Antarctic Data Centre, doi pending.

**Author Contribution**

CKC led the study, undertaking the majority of the data analysis and writing of the manuscript. TRV and ADF conceived the concept and design of the study, and contributed substantially (along with ARK) to writing the manuscript. TRV, NJA, MAJC, CKC, ASC, VF, AJEG, HAK, ADM, CTP, LMJ, PTV, JW and LZ contributed to the Mount Brown South ice core drilling project, ice core analysis and dating, calculation of snow accumulation rates and ice core site climatology. ARK and LMJ assisted in the plotting and interpretation of the near-surface wind vectors, SST, GPH and zonal wind composite anomaly
maps. All authors contributed to writing the manuscript, and declare that they have no conflict of interest.

**Acknowledgements**

This work was supported by ARC Discovery Project DP180102522, Australian Antarctic Science (AAS) projects 4414 and 4537, the ARC Special Research Initiative for Antarctic Gateway Partnership (SR140300001), the Australian Antarctic Program Partnership (ASCI000002), the Carlsberg Foundation, and contributes to US NSF project P2C2 18041212.




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
