# Peer review of "El Niño Southern Oscillation signal in a new East Antarctic ice core, Mount Brown South"

_Climate of the Past, 2020_

## Referee Comment (RC1) · Anonymous Referee #1 · 1 Dec 2020

In this manuscript, the authors used few short firn cores from an interesting coastal region in the Indian Ocean sector of the East Antarctica to identify the potential climatic records from this region through comparison with reanalysis data during 1975 - 2016. The study also attempts to compare and differentiate the climate archives at Mount Brown South with a well-known and extensively studied Law Dome site. Study of the Antarctic climate for the past millennia at seasonal to annual resolution using multiple records is a critical requirement for an improved understanding the natural variability and the recent human impact on Antarctic climate. Considering that only few ice core records are available with such time-resolution in the East Antarctica, recovering new ice core records from coastal sites with high snow accumulation rates are important. Therefore, the present study using short firn cores to identify the dominant climatic

signature embedded in the ice core archives at the Mount Brown South (MBS) region is a useful background study for a long-term climate reconstruction. However, the study needs to be more refined and revised to be suitable for publication.

Major comments: 1. Although introduction states that the study aims investigate potential signals for ENSO, SAM and IOD, and also section 3.3 mentions about the SAM and IOD, there exist no proper discussion and the whole effort looks like a half-hearted attempt. Mostly importantly, while there section 4.3 dealing with the lack of SAM record at MBS, the IOD component is completely missing other than stating that MBS salt record is not significantly correlated with DMI. Even if there exist no statistical correlation, it is important to discuss the details, potential reasons and substantiate that the site is reliable for reconstructing only ENSO. 2. One of the critical factors that influence the annual/seasonal snow accumulation rates in coastal Antarctica is the impact of extreme precipitation events (Turner et al., 2019, GRL). The coastal region around the East Antarctica studied here has shown to be strongly influenced by such events. This could also have significant influence on the seasonality of proxy records especially in high accumulation sites. Since impact of the climatic modes are season-dependent, it is important to have an analysis of potential impact of EPEs on the seasonal/annual climate record at the MBS. Undertaking an analysis of the precipitation and impact of extreme events using the updated high-resolution RACMO output could be very helpful. RACMO model is also considered to be more consistent on a long‐ term basis and also its performance has been extensively evaluated. Since the present study itself deals only the last some decades when reanalysis and model outputs are available (and reliable), it is important to add value to the palaeoclimatic perspective of this study by examining the impact of such local/ regional events. 3. Lack of proper tagging of figures in the methods and results are making it very hard to follow the data and analysis. Another issue that needs to be carefully revised is the mixing of methods in results and vice versa. Also some part of discussion is mixed up in the results section. A careful editing is warranted. 4. There are too many abbreviations in the manuscript making it tedious to follow. While it is acceptable to have the common acronyms like

ENSO, SAM, IOD, as well as shortening some of the most commonly used names (like MBS), the tendency to use acronyms for all and sundry should be avoided. Acronyms like MOCV, RWT, etc. are unnecessary and needs to be avoided.

Specific comments: Abstract L25 – Throughout, authors have used the term "snowfall accumulation rates". This is misleading, as the snow accumulation at a given site in Antarctica is a product of various processes other than just snowfall. This is especially true in coastal Antarctica as wind-induced redistribution are dominant as well as processes like diffusion are very common. Therefore, "snow accumulation rate" is a more correct term. L30 – Please explain and detail ". . .. suggesting occurrence of distinct moisture and aerosol intrusions". Such a sweeping statement without supporting evidence in the discussion doesn't help the discussion. Introduction L39 – Most of Antarctic ice cores are resolved at decadal or century scale, not millennial. L47 – Either the full form or just RICE. L49 – Such context is inappropriate; this study deals with only very short cores representing less than 40 years long. It is important to give the importance of array of cores for background of seasonal/annual records. Therefore, introduction may be revised to discuss more on records are available across Dronning Maud Land to Law Dome and beyond that would have more relevance on the science discussed here. L55- Avoid unnecessary acronyms that reduce the flow or reading. L74 – This line is confusing and has no relevance for this study. L98 – "Signals for ENSO in East Antarctica are more muted. . .". Be specific. East Antarctica is too large a place to make such sweeping statement. L101 – There are some recent studies on the influence of IOD on southern hemisphere and vice versa (Nuncio and Yuan, 2015, Journal of Climate; Zhang et al., 2020, GRL). It would be more interesting and valuable to look at the impact of these possibilities at MBS in discussion and find potential links. L134 – "Main"? This term is only explained later at Methods. Either define here or avoid using it. L137 – Revise. There are many high-resolution (seasonal/annual) ice core records that represent past 100 - 200 years of climate across the coastal East Antarctica. Therefore, there is nothing much to prove on Hypothesis 1. Also Vance et al. (2016) has also given such higher accumulation at this core site. Methods L141 –

[Figure]

Better to give the short forms in the title (MBS, LD) for helping the reader. L151 – "The MBS...". You mean the "Main"? L157 – Fig 1 is uninteresting and a missed opportunity to give more useful information. It would be very useful to give a schematic diagram of dominant features of ENSO/PSA impact around the Indian Ocean sector. L194 – See previous comments on the "snowfall accumulation rates". L203 – May replace "... the usual proxy..." with something like "the more conservative proxy.." L211 – A good part of this section (and methods in general) deals with results that could be best placed at results section. L225 – This needs an explanation in the context of the present study. Why summer for LD and annual for MBS? In Fig. 3, for accumulation, annual rates are used, but for sea salt it is summer. Such convenient picking needs to avoided or a more specific reasoning. L227 – As mentioned earlier, you need to refer the figures and tables as you start discussing. In absence of it, it is very difficult to follow the discussion. This is the case at many places. L244 – This does not explain why only Sept-Oct data of Law Dome was used for statistical study. Is there any data/reasoning to support that the ENSO is impacting MBS and LDS at different seasons? L267 – This is confusing. Revise. Results L279 – Section 3.1 title doesn't convey much. Why didn't you give a title that reflects what is discussed in the section? L281 – This entire para deals with chronological constraints that could be best placed at section 2.3. This section should deal with more on results of the study on proxy data. L319 – "... seasonal...". You need to give in bracket, which seasons for clarity. Also this para should give some explanation why JJASON for MBS and SO for LD records were used. Otherwise it is more an act of convenience. L326 – As commented earlier, it is important to refer to the Figures /Tables to guide the readers. L342 – The scale on Fig 7 (also Fig 6) needs check. Seems the sign missing. L360 – This section needs revision, as there is no discussion on the results on IOD. Also the data/figures are not referred. L363 – It is not correct that there is a "lack of a SAM signal". May consider to revise it as "lack of a statistically significant SAM ...." Discussion L377 – Section 4.1 title doesn't convey its content. May revise. L415 – As commented earlier, it is important to explore the impact of extreme precipitation events on the proxy records discussed here. Such an

evaluation would enhance its value for a journal like CP. L419 – The entire discussion on IPO is pointless as the records discussed here are less than 40 years! It's all speculation and lacks purpose. L429 – Some discussion on the potential mechanisms on the influence of ENSO to the study site is important. There are some previous studies on these that could be used as a starting framework. L450 – A discussion on the potential robustness of MBS records (compared to the LD site) for ENSO reconstruction would be useful. L459 – Exactly. This potential impact of extreme events needs to be explored. L467 – This is more speculation in the absence of any proof on "anomalies develop in austral winter than spring". Either you need to provide a proof or remove such statements. L471 – This is speculative at this stage without discussing proof and reasoning. L473 – Section 4.3 may include a discussion on IOD signal (or lack of it) at the MBS site. L475 – Is this consistent with the Marshall et al. (2017) study? If not, may be some reasoning needs to be brought out. Conclusion L520 – Section 5.0 needs revision. It also needs to be nuanced that it's a composite of 3 records. L526 – This doesn't actually reflect the important findings of the study; for example, the MBS is ideally suited for ENSO reconstruction and issues with SAM and IOD at this region.

---

## Referee Comment (RC2) · Anonymous Referee #2 · 29 Jan 2021

In the study by Crockart et al, the accumulation rate records and the chloride records of 3 shallow firn cores from Mount Brown South as well as from a new shallow core from Law Dome are presented. The correlation of these records to the El Nino Southern Oscillation signal is investigated by means of Pearssons correlation coefficient and discussed. The aim of the study is to show, that a future longer record from the site will contain climate variability signatures such as the Southern Oscillation and will give different/additional information than a comparable record from Law Dome.

Obtaining and analyzing ice cores from coastal Antarctica is highly valuable in order to resolve sub-annual and annual climate variability. However, the presented study aims to look at variability at larger time scales in order to evaluate the benefit of a future longer ice core record. This overall aim prevents a thoughtful analysis of the certainly

valuable data available.

The paper would benefit from a clear and consistent outline on - how the cores and records are obtained - a clear presentation of the full data set - a clear outline on which time scales are investigated and why (by which species, resolution –which processing step) and what possible drawbacks/bias have to be considered, i.e. Example 1: comparing annual values of accumulation rate – by averaging the annual accumulation rates of all the three firn cores (arithmetic average?), problem: lack of high-res density – what is the influence of intrusion, single events on the overall record/correlation analysis; Example 2: looking over a time period of 40 years (in comparison to ERA5 etc)-which is by no means close to centennial. What is the significance of this correlation over this time period with respect to a future millennial record? Did you test for trends (affecting the correlation?) What if the correlation is (only) a result of recent changes and not due to variability itself? For variability analysis the trends have to be looked at (and removed)? - a better structure in distinguishing between results and discussion - a focus on the available time scale, the high-resolution data available and what one can learn from it

Further general remarks: 1. Terms on climate variability time scales are used inflationary and ambiguous throughout the text, for example: High-resolution ice core – long term climate variability and multi-centennial – all in one sentence/context. (Abstract, Line 21-22, Introduction Line 44), Usage of very different time scales (as relevant for the paper), for example: variability over past millennia (Intro, Line 55), sub-decadal (?) signals for climate variability (Intro, Line 126), multi-centennial (Intro, Line 86) Suggestion: There should be a clear outline of which times scales the authors aim to address with their data – both in resolution (sub-annual) and coverage (30-40 years!). It is in the nature of coastal high-res records, that they resolve-sub-annually ( a benefit which is unfortunately not discussed or presented here more in-depth). Overall the (only) time scale considered here is annual mean over a period of 40 years – for both accumulation rate and sea salt concentration.
2. Talking about climate signals/climate variability at different time scales would require a profound analysis of the climate signal contained in the record (especially given the fact, that overall a short period of 40 years are considered. What is the common climate signal in the three cores? And what the common climate signal with the core from Law Dome? Are the 40 years long enough? What about variability vs mean/trend?

3. Already in the introduction 14 different abbreviations are introduced (and used later in the text) – its hard to read and to follow your argumentation. Maybe it is possible to stick to few, relevant terms – as not all of the modes are used later or relevant for the paper (or one could stick to summarized, overarching terms of comparable modes). 4. Information to the firn cores Do you analyze the core until the surface? How do you deal with the upper meter(s). How do you cut the core in the field, transportation, cutting in the lab... A table of the exact coordinates, the length/logging depth etc is missing, and then the obtained coverage in time There is inconsistency in the number of cores included in your analysis. In chapter 2.1 (methods) it reads, that from the three MBS firn cores, the Bravo core is not used in the study. That makes 2 shallow cores plus the upper part of the main core. i.e three cores from the MBS site. However in line 133 it reads: The MBS record is unique in that it contains three short ice cores (20-25m) (...) in addition to the Main core. There is also unclear usage of the term: "record" – is this always meant as the stacked (averaged) record over the three (?) cores? This should be stated clearly.

Comments in detail: Line 21: wording in combination: high-res ice core vs long term, multi-centennial versus long-term Line 30: occurrence of moisture and aerosol intrusions -is breefly touched in the discussion but not shown in the data Line 52: high-resolution records are required to fill spatial gaps... High-res records address temporal information, not spatial? Line 141: please explain, what is meant by "wet deposition"? Line 151: The MBS record: It is not explained, how you derive the "record" here – what exactly is done? The record of the single accumulation rates or the stacked/merged? Line 160-165: In the figures only 1 record for LD is shown – how did you combine
the old and new records? Where they exactly the same (in the common period)? A little note on how the two records are combined is needed here Chapter 2.4. Deriving accumulation rates from empirical density model? Are there now bag mean densities obtained in the field/lab? As accumulation rate is one major result of the study and it is based on the empirical equation, it would be good to show the density data (modeled in comparison to bag means). In any case, different layers of density will not be considered and may bias the derived ice equivalents. A more in-depth description/uncertainty analysis should be given (again, based on the fact, that this is one of major results of the paper). Line 198-199: I do not understand: You convert your profile to ice equivalent in order to do exactly this: to compare different layers of different depth of an ice core, no matter of the thinning (by compaction). If you refer to thinning because of flow then it reads very strange, given the fact, that you are looking at 25m depth max.

Equation 1: description/ labeling of the terms is missing (i.e d = depth, what are the number standing for?) Line 280: The MBS record... Again, it is not clear to what is referred here. Figure2: why do you not show the full record? Line 302/303: wind/blown snow effect- where has this been shown? (Reference or short explanation how)

**CPD**

---

## Author Response (AR1)

Response to review comments on the paper 'El Niño Southern Oscillation signal in a new East Antarctic ice core, Mount Brown South' by Crockart et al. (manuscript number cp-2020-134).

**Reply to comments from Reviewer #1.**

In this manuscript, the authors used few short firn cores from an interesting coastal region in the Indian Ocean sector of the East Antarctica to identify the potential climatic records from this region through comparison with reanalysis data during 1975 - 2016. The study also attempts to compare and differentiate the climate archives at Mount Brown South with a well-known and extensively studied Law Dome site. Study of the Antarctic climate for the past millennia at seasonal to annual resolution using multiple records is a critical requirement for an improved understanding the natural variability and the recent human impact on Antarctic climate. Considering that only few ice core records are available with such time-resolution in the East Antarctica, recovering new ice core records from coastal sites with high snow accumulation rates are important. Therefore, the present study using short firn cores to identify the dominant climatic signature embedded in the ice core archives at the Mount Brown South (MBS) region is a useful background study for a long-term climate reconstruction. However, the study needs to be more refined and revised to be suitable for publication.

> **Author response:** Thank you for your valuable comments and suggestions. We will revise the paper and make amendments. This is my (Camilla Crockart) first paper and is the publication from my MSc project. I will now be developing the longer MBS record in my PhD.

> **Author changes:** Thank you for your valuable comments and suggestions. We have revised the paper and made amendments.

Major comments: 1. Although introduction states that the study aims investigate potential signals for ENSO, SAM and IOD, and also section 3.3 mentions about the SAM and IOD, there exist no proper discussion and the whole effort looks like a half-hearted attempt. Mostly importantly, while there section 4.3 dealing with the lack of SAM record at MBS, the IOD component is completely missing other than stating that MBS salt record is not significantly correlated with DMI. Even if there exist no statistical correlation, it is important to discuss the details, potential reasons and substantiate that the site is reliable for reconstructing only ENSO.

> **Author response:** Adding more detail on the lack of a statically significant IOD signal is a good idea. We will add a section that discusses potential reasons for the lack of statistically significant IOD and SAM signals in the MBS record (Section 4.3). As the SAM varies from weekly to seasonal timescales, it is possible that a statically significant SAM signal is only preserved in certain months or seasons in the MBS record, such as in the case of LD (Vance et al. 2013). Similarly, the IOD is seasonally locked - peaking in spring - and therefore an IOD signal, if preserved, may only be preserved in the seasonal MBS record, potentially at a resolution that we may not yet be able to resolve. As the frequency of precipitation is yet to be determined in detail, we think it is important to keep the MBS record annually resolved at this stage to avoid any errors associated with binning monthly values. Therefore, potential SAM and IOD signals preserved in the seasonal MBS record will need to be examined in the future, once precipitation at MBS has been investigated in more detail. We would like to stress that this is the first analysis of annual signals in the new MBS ice cores. Future work is ongoing, and we will endeavour to make clearer in the manuscript that study on signals derived from the longer Main MBS datasets is ongoing.

> **Author changes:** We have added a paragraph in the Discussion (starts L516) that discusses why there might not be a IOD signal in the annual MBS record presented here.

2. One of the critical factors that influence the annual/seasonal snow accumulation rates in coastal Antarctica is the impact of extreme precipitation events (Turner et al., 2019, GRL). The coastal region around the East Antarctica studied here has shown to be strongly influenced by such events. This could also have significant influence on the seasonality of proxy records especially in high accumulation sites. Since impact of the climatic modes are season-dependent, it is important to have an analysis of potential impact of EPEs on the seasonal/annual climate record at the MBS. Undertaking an analysis of the precipitation and impact of extreme events using the updated high-resolution RACMO output could be very helpful. RACMO model is also considered to be more consistent on a longer term basis and also its performance has been extensively evaluated. Since the present study itself deals only the last some decades when reanalysis and model outputs are available (and reliable), it is important to add value to the palaeoclimatic perspective of this study by examining the impact of such local/ regional events.

**Author response:** That's a good idea. Given the co-authors we have on this work (e.g., Vincent Favier and Jonathan Wille) and their expertise in the French Modele Atomspherique Regionale (MAR), we will add additional analysis using the MAR model instead of RACMO. Agosta et al. (2019) suggest that the MAR and RACMO perform similarly well in simulating surface mass balance gradients in both plateau and coastal regions of Antarctica. With the help of a new co-author Christoph Kittel (as well as Favier and Wille), we propose to add additional analysis using the surface mass balance data from the MAR to look at the frequency of precipitation at MBS at a monthly/seasonal scale.

**Author changes:** We have added additional analysis using the surface mass balance variable from the MAR model, instead of RACMO. Also, Vance et al. (2016) looked at RACMO and showed that it underestimated surface mass balance in the MBS region. We have added analysis looking at the climatology, seasonal variability and extremes in surface mass balance at MBS. Please see Fig. 4, and L462 in Methods, L518 in Methods, and L701, L796, L871 in Discussion.

3. Lack of proper tagging of figures in the methods and results are making it very hard to follow the data and analysis. Another issue that needs to be carefully revised is the mixing of methods in results and vice versa. Also some part of discussion is mixed up in the results section. A careful editing is warranted.

**Author response:** Apologies. This will be carefully edited so that there is less overlap between the methods, results and discussion. All figures and tables will be correctly labelled and appropriately referred to in the results and discussion sections.

**Author changes:** This has been carefully edited so that there is less overlap between the methods, results and discussion.

4. There are too many abbreviations in the manuscript making it tedious to follow. While it is acceptable to have the common acronyms like ENSO, SAM, IOD, as well as shortening some of the most commonly used names (like MBS), the tendency to use acronyms for all and sundry should be avoided. Acronyms like MOCV, RWT, etc. are unnecessary and needs to be avoided.

**Author response:** We will revise using only standard acronyms (e.g., MBS, LD, IOD, ENSO, and SAM).

**Author changes:** We have revised using only standard acronyms (e.g., MBS, LD, IOD, ENSO, and SAM).

Specific comments: Abstract L25 – Throughout, author have used the term "snowfall accumulation rates". This is misleading, as the snow accumulation at a given site in Antarctica is a product of various processes other than just snowfall. This is especially true in coastal Antarctica as wind-induced redistribution are dominant as well as processes like diffusion are very common. Therefore, "snow accumulation rate" is a more correct term.

**Author response:** This will be corrected to 'snow accumulation rate' throughout the paper.

**Author changes:** This has been corrected to *'snow accumulation rate'* throughout the paper.

L30 – Please explain and detail ". . .. suggesting occurrence of distinct moisture and aerosol intrusions". Such a sweeping statement without supporting evidence in the discussion doesn't help the discussion.

**Author response:** We will revise the statement to be less sweeping and suggest that 'further study is likely to help us define the cause of this inverse relationship between LD-ENSO and MBS-ENSO'. The first piece of evidence to help with this analysis is the work of Udy et al. 2021, which showed that the two ice core sites have differing moisture sources at the synoptic scale, and our results support this finding. Please note it was a condition of the MBS site selection that the MBS record differed to the LD record (Vance et al. 2016). Our results support this as the two records differ (e.g., the annual sea salt concentrations are not correlated, and the MBS-annual and LD-summer sea salts are inversely correlated).

**Author changes:** We have written '*The Mount Brown South site record and Law Dome record preserve inverse signals for the ENSO, possible due to longitudinal variability in meridional transport in the southern Indian Ocean, although further analysis is needed to confirm this*'. See L28.

Introduction L39 – Most of Antarctic ice cores are resolved at decadal or century scale, not millennial.

**Author response:** This will be corrected to 'centennial'.

**Author changes:** This has been corrected to '*decadally to centennially-resolved*'.

L47 – Either the full form or just RICE.

**Author response:** This will be corrected to the full form, 'Roosevelt Island Climate Evolution'.

**Author changes:** This has been removed from the manuscript.

L49 – Such context is inappropriate; this study deals with only very short cores representing less than 40 years long. It is important to give the importance of array of cores for background of seasonal/annual records. Therefore, introduction may be revised to discuss more on records are available across Dronning Maud Land to Law Dome and beyond that would have more relevance on the science discussed here.

**Author response:** We will make it clear that we are only using the upper section of the MBS ice core. We propose to add more detail on high-resolution ice cores collected from East Antarctica, including Dronning Maud Land, and remove any of the West Antarctic examples in a marked-up version of the manuscript.

**Author changes:** We have specified that we are looking at the Indian Ocean sector of Antarctica, and therefore have removed any West Antarctica examples and East Antarctica examples that are not in the Indian Ocean sector of Antarctica. See L46.

L55- Avoid unnecessary acronyms that reduce the flow or reading.

**Author response:** As above – we will revise using only standard acronyms.

**Author changes:** We have revised using only standard acronyms.

L74 – This line is confusing and has no relevance for this study.

**Author response:** This will be corrected to 'Vance et al. (2016) suggest that a new ice core collected from the MBS region may contain an independent SAM signal'.

**Author changes:** This has been corrected to '*Vance et al. (2016) suggest that a new ice core collected from the MBS region may contain an independent SAM signal*'. See L70.

L98 – "Signals for ENSO in East Antarctica are more muted. . .". Be specific. East Antarctica is too large a place to make such sweeping statement.

160

**Author response:** This will be corrected to 'The ENSO also influences East Antarctica, for example, there is an ENSO signal preserved in the summer sea salt record from LD, although there are far fewer studies on ENSO influences at high latitudes in the East Antarctic to determine this definitively (Vance et al. 2013)'.

**Author changes:** This has been corrected to *'The ENSO also influences East Antarctica, for example, there is an ENSO signal preserved in the summer sea salt record from LD (Vance et al. 2013)'*. See L95.

165

L101 – There are some recent studies on the influence of IOD on southern hemisphere and vice versa (Nuncio and Yuan, 2015, Journal of Climate; Zhang et al., 2020, GRL). It would be more interesting and valuable to look at the impact of these possibilities at MBS in discussion and find potential links.

170

**Author response:** We will add more detail on the influence of the IOD in Antarctica and refer to the above-mentioned papers. As there is limited literature on the influence of the IOD in East Antarctica, particularly the Indian Ocean sector, we will add more detail in the discussion section explaining the lack of information and data on the influence of the IOD in East Antarctica.

175

**Author changes:** We have added a paragraph in the Discussion (see L517) that discusses why we do not find a IOD signal in the annual MBS records presented here.

L134 – "Main"? This term is only explained later at Methods. Either define here or avoid using it.

180

**Author response:** This will be changed to the 'the extended MBS Main ice core'.

**Author changes:** This has been changed to *'a long core'*.

L137 – Revise. There are many high-resolution (seasonal/annual) ice core records that represent past 100 - 200 years of climate across the coastal East Antarctica. Therefore, there is nothing much to prove on Hypothesis 1. Also Vance et al. (2016) has also given such higher accumulation at this core site.

**Author response:** This will be change to '(Hypothesis 1) contains signals for past climate variability at a high-resolution that extend beyond short spatial scale variability, and (Hypothesis 2) contains climate signals that differ from the LD record.' We would like to determine whether the MBS record preserves climate signals from the lower latitudes (i.e., outside of the Antarctic region), and whether it meets the drilling requirements of Vance et al. (2016) of containing a complementary ice core record to LD. Although we present the short MBS record here, we think it is important to introduce the longer record that spans 1,000 years. The brevity and scarcity of high-resolution ice cores in this region is a limitation of our understanding of climate variability (Jones et al. 2016; Stenni et al. 2017; Vance et al. 2016). Hence, both the length and location of the extended Main MBS record is what makes this ice core record unique. We would like to emphasise this point, while making it clear that the extended Main ice core has not yet been fully developed. Additionally, the upper portion of the Main MBS core constitutes part of this analysis. We disagree that there are 'numerous high-resolution cores spanning 100-200 years in East Antarctica'. Yes, there are more in Dronning Maud Land, but in the Indian ocean sector from Enderby Land through to Wilkes Land there are very few. We propose to change this to define as the 'Indian Ocean sector from Enderby to Wilkes Land'. This is still a vast section of coastline in East Antarctica.

**Author changes:** This has been changed to *'(Hypothesis 1) contains signals for past climate variability at a high-resolution that extend beyond short spatial scale variability, and (Hypothesis 2) contains climate signals that differ from the LD record.'* See above for explanation.

Methods L141 – C3 Better to give the short forms in the title (MBS, LD) for helping the reader.

210

**Author response:** Good suggestion. This will be changed to 'MBS and LD (Dome Summit South site)'

**Author changes:** This has been changed to *'MBS and LD (Dome Summit South site)'*. See L138.

L151 – "The MBS. . .". You mean the "Main"?

**Author response:** That's correct. This will be corrected to the 'Main'.

**Author changes:** This has been changed to *'Main'*.

L157 – Fig 1 is uninteresting and a missed opportunity to give more useful information. It would be very useful to give a schematic diagram of dominant features of ENSO/PSA impact around the Indian Ocean sector.

**Author response:** We agree that Figure 1 could have contained more detail. However, we would prefer to have a site map in the methods section detailing the exact layout of the new ice core site where the MBS ice cores were drilled. We will change Figure 1 to include a map of the four ice core drilling sites at MBS, as the second Reviewer has asked for more detail on the MBS drilling site. Again, this is the first data paper for the new MBS ice core/s, and we think a detailed site map is appropriate (e.g., similar to Abram et al., 2011).

**Author changes:** We have changed Fig. 1 one to be a more detailed site map, with the longitude and latitude of the individual MBS ice cores.

L194 – See previous comments on the "snowfall accumulation rates".

**Author response:** This will be corrected to 'snow accumulation rate' throughout the paper.

**Author changes:** This has been corrected to *'snow accumulation rate'* throughout the paper.

L203 – May replace ". . . the usual proxy. . ." with something like "the more conservative proxy.."

**Author response:** This will be changed to 'the more conservative proxy'.

**Author changes:** This has been corrected to *'the more conservative proxy'*.

L211 – A good part of this section (and methods in general) deals with results that could be best placed at results section.

**Author response:** Apologies for the overlap between the methods and results. This will be carefully edited in the marked-up version.

**Author changes:** This has been carefully edited in the marked-up version.

L225 – This needs an explanation in the context of the present study. Why summer for LD and annual for MBS?

**Author response:** We will add a more thorough explanation so that it is clearer for the reader. We focused on the annual MBS record, as detailed analysis of the frequency of precipitation at MBS is the subject of extensive future study (e.g., with Favier and Wille), meaning it is not appropriate yet to use a seasonal MBS record due to errors associated with interpolating monthly values when the uniformity (or otherwise) of annual accumulation has not yet been properly assessed. As the frequency of precipitation at LD has been extensively studied, and it is known that extreme precipitation events have little impact on annual accumulation at LD, it is appropriate to use seasonal LD records (e.g., van Ommen & Morgan, 1997, McMorrow et al. 2001; 2002, Pedro et al., 2011; Vance et al

260  2013; 2015, Roberts et al., 2015). We chose to focus on the summer sea salt record from LD for the composite analysis because it is known to preserve a signal for the ENSO (e.g., Vance et al. 2013).

**Author changes:** Please see L222. Note that we have used December-March instead of the standard summer season to remain consistent with Vance et al. (2013).

265  In Fig. 3, for accumulation, annual rates are used, but for sea salt it is summer. Such convenient picking needs to avoided or a more specific reasoning.

**Author response:** The LD sea salt and annual accumulation records are established climate proxies containing climate signals as diverse as modes of variability (e.g., ENSO, IPO, SAM) and surface climate (Australian rainfall, 270  south-west Pacific circulation variability) (e.g., Goodwin et al., 2004; van Ommen & Morgan, 2010; Vance et al. 2013; 2015; Roberts et al., 2015; 2019; Marshall et al., 2017; Udy et al., 2021), and hence we disagree that these are convenient picks. In the original manuscript we used a standard seasonal average for the LD summer sea salts (December-February). Given the Reviewer 1's concerns, we propose to use a December-March average salt concentration as the proxy to test for LD links to ENSO, as this is the established, published proxy for the ENSO 275  and eastern Australian rainfall from Law Dome (Vance et al. 2013; 2015). This means we remain consistent with previous studies which have been developed in detail including analyses of the mechanisms responsible for the summer signals preserved at Law Dome (Vance et al., 2013; Udy et al., 2021; Udy et al., in prep), which may allay the concerns of the reviewer. Note that a change from a December-February sea salt average to a December-March sea salt average produces essentially the same timeseries ($r = 0.911$, $p = 0.000$). As a result, the findings 280  in the paper are identical. For the new MBS ice cores, this initial study looks only at annual resolution because we need to undertake further study on signal preservation prior to looking at seasonally resolved records.

**Author changes:** See above.

285  L227 – As mentioned earlier, you need to refer the figures and tables as you start discussing. In absence of it, it is very difficult to follow the discussion. This is the case at many places.

**Author response:** Apologies, all figures and tables will be correctly labelled and referred to in the results and discussion sections.
290
**Author changes:** All figures and tables are correctly labelled and referred to in the Results and Discussion sections.

L244 – This does not explain why only Sept-Oct data of Law Dome was used for statistical study. Is there any 295  data/reasoning to support that the ENSO is impacting MBS and LDS at different seasons?

**Author response:** The climate indices months correlated against the LD record were selected based on the findings of Vance et al. (2013), which used September-October averages of ENSO indices. We propose to use a standard seasonal average for austral spring (i.e., September-November) in order to simplify the story, but still 300  maintain consistency with the proxy records developed previously (i.e., December-March sea salt concentrations) This does not change the findings of the manuscript (e.g., the correlation between the LD summer sea salts and the ENSO indices are still significant, e.g., $r = 0.398$, $p = 0.01$ Southern Oscillation Index). We do not suggest that the ENSO impacts MBS and LD during different seasons, rather that the ENSO signal is preserved differently in the two ice core records, as these sites are ~1000 km apart, at different elevations and subject to different 305  synoptic scale processes. Udy et al. 2021 suggest differing moisture transport to the two ice cores sites, implying that any climate signals may be preserved differently at the two sites. One explanation for why the LD record preserves an ENSO signal in the summer may be because the LD record may be a noisier record in winter, and therefore any signals for the ENSO in winter may be masked by local weather systems (see L407). Future detailed analysis on precipitation at MBS is needed to determine whether this is the case, however, this is too large a task 310  for the present study and is the subject of ongoing and future work.

**Author changes:** We have used the standard spring season (September-November), instead of September-October used in Vance et al. (2013), please see L245.

315    L267 – This is confusing. Revise.

**Author response:** This will be changed to 'The composite years are based on anomalous sea salt years, while the months displayed were chosen based on months that had significant correlations with the relevant climate modes'.

320 **Author changes:** This has been changed to *'The composite years are based on anomalous sea salt years, while the months displayed were chosen based on months that had significant correlations with the relevant climate modes'.* See L268.

Results L279 – Section 3.1 title doesn't convey much. Why didn't you give a title that reflects what is discussed in the
325    section?

**Author response:** This will be changed to 'MBS ice core features 1975-2016'.

**Author changes:** This has been changed to *'MBS and LD dating, snow accumulation and sea salt concentrations'.*
330    See L291.

L281 – This entire para deals with chronological constraints that could be best placed at section 2.3. This section should
deal with more on results of the study on proxy data.

335 **Author response:** We agree to edit the methods, results and discussion sections so that there is less overlap. In regard to this paragraph, the second Reviewer argues that the main result of the paper is the accumulation and sea salt datasets (rather than any climate mode signals). We agree with the second Reviewer, as this is the first manuscript for this new ice core, therefore, we propose to leave this paragraph in the results. However, we will revise for clarity.
340

**Author changes:** See above.

L319 – ". . . seasonal. . .". You need to give in bracket, which seasons for clarity. Also this para should give some
explanation why JJASON for MBS and SO for LD records were used. Otherwise it is more an act of convenience.
345

**Author response:** This will be corrected to 'The MBS sea salt site record is significantly correlated with the seasonal Multivariate ENSO Index, Niño 4, Niño 3.4, and the Southern Oscillation Index (June-November, see Table 1)'. As above – the September-October averages of the ENSO indices were based on the Vance et al. (2013), although we propose to extend this to include September-November to make it a seasonal average. June-
350    November ENSO indices were correlated against the annual MBS site record because sea surface temperature anomalies and convection anomalies related to the ENSO tend to emerge in early austral winter in the equatorial and south-west Pacific and propagate to higher southern latitudes during austral spring and into summer (Fogt and Bromwich 2006; L'Heureux and Thompson 2006), and we are developing an annual record for MBS at this stage. This means, it makes more sense to test against the spread of seasons where the ENSO anomalies develop in the
355    Southern Hemisphere and high latitudes. In addition, we wished to align as closely as practicable to annual MBS sea salt record, and hence do not extend the ENSO indices into the following year. We have correlated the MBS sea salt record against the ENSO indices in May-December, June-December, May-November and the difference in *r*-values and *p*-values are negligible. Again, this is the first paper for MBS – there are currently multiple studies underway developing more understanding of the signals and mechanisms to deliver these signals preserved at
360    MBS.

**Author changes:** This has been corrected to *'The MBS sea salt site record is significantly correlated with the seasonal Multivariate ENSO Index, Niño 4, Niño 3.4, and the Southern Oscillation Index (June-November, see Table 1)'*. See L339.

365

L326 – As commented earlier, it is important to refer to the Figures /Tables to guide the readers.

**Author response:** Apologies, all figures and tables will be correctly labelled and appropriately referred to in the results and discussion sections.

370

**Author changes:** All figures and tables will be correctly labelled and appropriately referred to in the results and discussion sections.

L342 – The scale on Fig 7 (also Fig 6) needs check. Seems the sign missing.

375

**Author response:** Apologies, negative signs on the scalebar will be added.

**Author changes:** Negative signs on the scalebar have been added to Fig. 6 and Fig. 7.

380   L360 – This section needs revision, as there is no discussion on the results on IOD.

**Author response:** We appreciate this is the case however, there is very limited literature on the link between the IOD and high latitude climate, particularly in the Indian Ocean sector. We will add more detail in the discussion section explaining the lack of information and data on the influence of the IOD on the high latitudes (particularly the Indian Ocean sector), which may explain why we do not get a statistically significant IOD signal in the MBS

385   record. Moreover, the IOD is seasonally locked and may only be preserved, if preserved, in the seasonal MBS record, which has not been developed yet. Please note that a seasonal MBS record may only be developed after the uniformity (or otherwise) of annual precipitation at MBS is investigated thoroughly, which is ongoing work.

390   **Author changes:** See above, note there is a new paragraph discussing the lack of a IOD signal in the annual MBS records presented here (starts L517).

Also the data/figures are not referred.

395   **Author response:** Apologies, all figures will be referred to correctly and appropriately referred to in the results and discussion sections.

**Author changes:** All figures are to correctly and appropriately referred to in the Results and Discussion sections.

400   L363 – It is not correct that there is a "lack of a SAM signal". May consider to revise it as "lack of a statistically significant SAM . . .."

**Author response:** This will be changed to the 'lack of a statistically significant SAM signal'.

405   **Author changes:** This has been changed to *'lack of a statistically significant SAM signal'*.

Discussion L377 – Section 4.1 title doesn't convey its content. May revise.

**Author response:** We will revise to better suit the content once the results, methods and discussion sections have

410   been revised.

**Author changes:** This has been changed to '*MBS and LD ice core features 1975-2016'*. See L390.

L415 – As commented earlier, it is important to explore the impact of extreme precipitation events on the proxy records discussed here. Such an evaluation would enhance its value for a journal like CP.

> **Author response:** Determining the impact of extreme precipitation events at MBS in detail is a study in itself and will be investigated thoroughly in the near future. However, we propose to add additional analysis using the surface mass balance data from the Modèle Atmosphérique Régional to look at the frequency of precipitation at MBS at a monthly/seasonal scale.

> **Author changes:** We have added analysis that looks at the climatology of surface mass balance (using the MAR model) and precipitation (using ERA5) at MBS. Based on initial results here, extreme events occur in all seasons, and are not skewed to one season. Please see Fig. 4, and L462 in Methods, L518 in Methods, and L701, L796, L871 in Discussion.

L419 – The entire discussion on IPO is pointless as the records discussed here are less than 40 years! It's all speculation and lacks purpose.

> **Author response:** We disagree. In this study, we have not investigated the IPO as it requires a longer dataset. However, we do mention it in the discussion, as it is important to recognise that decadal variability may influence the strength of the ENSO signal in the MBS record, as it does influence the strength of ENSO signal in the LD record (Vance et al. 2013). There is strong decadal and multidecadal climate variability in Antarctica, which is often not captured in the observational data (Jones et al. 2016), and a highly cited reconstruction of the IPO has been developed form the LD record (Vance et al. 2015). Determining decadal climate variability is a key objective of the MBS ice core project (Vance et al. 2016), hence we think it is important to state that decadal variability may influence the strength and the stationarity of any climate signal preserved in the MBS ice core. Moreover, the second Reviewer argues that the ENSO signal may change over a longer timescale, therefore we think it is important to mention the potential causes of decadal variability in the ENSO signal (i.e., the IPO), and that we will be exploring decadal variability upon the development of the longer MBS datasets.

> **Author changes:** See above.

L429 – Some discussion on the potential mechanisms on the influence of ENSO to the study site is important. There are some previous studies on these that could be used as a starting framework.

> **Author response:** We propose to add more detail on mechanisms controlling the transport of ENSO signals from the equatorial Pacific to high-southern latitudes using previous studies (e.g., Turner et al., 2004; Vance et al. 2013, Fogt et al. 2006, Datwyler et al. 2020, and Clem et al. 2019). Figure 8 and the discussion around this figure is our first attempt at describing the mechanism linking the broader ENSO variability to winds in the southern Indian Ocean, and thereby to variability in sea salt aerosol generation and deposition at MBS. We will make this clearer in the manuscript and link it to previous work.

> **Author changes:** We have added more detail on the potential mechanisms for the ENSO signal at MBS, please see paragraph starting at L450.

L450 – A discussion on the potential robustness of MBS records (compared to the LD site) for ENSO reconstruction would be useful.

> **Author response:** Given the short record, we suggest there is some evidence for a robust ENSO signal. However, one of the benefits to studying ENSO in palaeoclimatology is the long-observed records of sea surface temperature spanning up to 150 years. As we continue analysing and dating the MBS record back in time, we will be able to determine with more confidence not only the robustness of the signal, but also any decadal variability (e.g., from the interaction of the IPO) or whether there is a stationary or non-stationary aspect to the signal. We will highlight this in more detail in the conclusion.

**Author changes:** Please see Section 4.4 (Robustness of the ENSO signal in the MBS record).

L459 – Exactly. This potential impact of extreme events needs to be explored.

**Author response:** As above - determining the impact of extreme precipitation events at MBS will be investigated thoroughly in the near future, as it is a large task and is likely a whole paper in itself. However, we propose to add extra analysis using the surface mass balance data from the Modèle Atmosphérique Régional to look at the frequency of precipitation at MBS at a monthly/seasonal timescale.

**Author changes:** Our initial results using the MAR dataset suggests that surface mass balance is relatively uniform throughout the year, and extreme events occur in all seasons. Please see Fig. 4, and L462 in Methods, L518 in Methods, and L701, L796, L871 in Discussion.

L467 – This is more speculation in the absence of any proof on "anomalies develop in austral winter than spring". Either you need to provide a proof or remove such statements.

**Author response:** This is based on Figure 8, as the winter anomalies are stronger and more extensive than the spring anomalies. However, we agree that this was not written very clearly. It will be revised.

**Author changes:** See above, also analysis of the MAR dataset suggest that surface mass balance is higher in winter compared to spring, hence there are stronger anomalies in winter compared to spring (see L457).

L471 – This is speculative at this stage without discussing proof and reasoning.

**Author response:** This will be revised to 'It is possible that particular off-shore circulation features shown in Fig. 8c and 8e are influenced by a teleconnection related to the ENSO'.

**Author changes:** This has been changed to *'Further analysis is required to determine how these teleconnections resulting from the ENSO manifest in the southern Indian Ocean north of the MBS site'*. See L462.

L473 – Section 4.3 may include a discussion on IOD signal (or lack of it) at the MBS site.

**Author response:** We will add more detail in the discussion section explaining the lack of information and data on the influence of the IOD on the high southern latitudes (particularly the Indian Ocean sector), which will help to explain why we did not find a statistically significant IOD signal in the MBS record. Moreover, the IOD is seasonally locked meaning that an IOD signal, if preserved, may only be preserved in the seasonal MBS record, which has not yet been developed.

**Author changes:** Please see revised Section 4.3, specifically the paragraph starting on L517.

L475 – Is this consistent with the Marshall et al. (2017) study? If not, may be some reasoning needs to be brought out.

**Author response:** Apologies, we are unclear why Marshall et al. 2017 would apply here. Marshall et al. 2017 discussed evidence of SAM signals at LD compared to Byrd (West Antarctica) ice cores in annual accumulation. We maintain that we will have to produce longer, and seasonally resolved records prior to definitively investigating whether there is a SAM signal preserved at MBS.

**Author changes:** See above.

Conclusion L520 – Section 5.0 needs revision. It also needs to be nuanced that it's a composite of 3 records.

520

**Author response:** The conclusion will be revised. When referring to the site average from the three MBS records, we will write the 'MBS site record' to avoid confusion. This was also noted by the second Reviewer. We will state the fact that we only use the upper section of the MBS ice core, while still mentioning that there is a longer record coming soon.

**Author changes:** Please see revised conclusion.

525 L526 – This doesn't actually reflect the important findings of the study; for example, the MBS is ideally suited for ENSO reconstruction and issues with SAM and IOD at this region.

**Author response:** Please note that the second Reviewer has asked for more emphasis on the accumulation and sea salt datasets, rather than any climate mode signals. Therefore, we will revise the conclusion and try to balance 530 the emphasis on both findings.

**Author changes:** As above, please see revised conclusion.

**Reply to comments from Reviewer #2.**
535

In the study by Crockart et al, the accumulation rate records and the chloride records of 3 shallow firn cores from Mount Brown South as well as from a new shallow core from Law Dome are presented. The correlation of these records to the El Nino Southern Oscillation signal is investigated by means of Pearsons correlation coefficient and discussed. The aim of the study is to show, that a future longer record from the site will contain climate variability signatures such as the Southern 540 Oscillation and will give different/additional information than a comparable record from Law Dome. Obtaining and analyzing ice cores from coastal Antarctica is highly valuable in order to resolve sub-annual and annual climate variability. However, the presented study aims to look at variability at larger time scales in order to evaluate the benefit of a future longer ice core record. This overall aim prevents a thoughtful analysis of the certainly valuable data available. The paper would benefit from a clear and consistent outline on - how the cores and records are obtained – a clear presentation of the 545 full data set – a clear outline on which time scales are investigated and why (by which species, resolution –which processing step)

**Author response:** Thank you for your valuable comments and suggestions. We will revise the paper and make amendments. This is my (Camilla Crockart) first paper and is the publication from my MSc project. I will now be 550 developing the longer MBS record in my PhD. Our aim for this manuscript is to determine whether the MBS record preserves climate signals from the lower latitudes (i.e., climate signals outside of the Antarctic region), and whether it meets the drilling requirements of Vance et al. (2016) of containing a complementary ice core record to LD. Although we present the short MBS record here, we think it is important to introduce the longer record that spans 1,000 years. The brevity and scarcity of high-resolution ice cores in Antarctica is a limitation of our 555 understanding of climate variability (Jones et al. 2016; Stenni et al. 2017; Vance et al. 2016). Hence, both the length and location of the extended Main MBS record is what makes this ice core record unique. We would like to emphasise this point, while making it clear that the extended Main MBS ice core has not yet been fully developed. Additionally, the upper portion of the Main MBS core constitutes part of this analysis. We will add more detail on how the cores were obtained in the field and how they were transported. In terms of the chemical 560 species, section 2.2 reads 'In order to investigate the optimal sample resolution over the satellite era for accurate dating, we cut 1.5 cm isotope samples in contrast to 3 cm chemistry samples over the upper portion of the ice cores'. And 'The Thermo-Fisher/Dionex ICS3000 ion chromatograph was used to determine the concentrations of trace ion chemistry (anions and cations), including sea salt concentrations (chloride ($Cl^-$), sodium ($Na^+$), magnesium ($Mg^{2+}$)), calcium ($Ca^{2+}$) and sulphate ($SO_4^{2-}$), as well as methanesulfonic acid (MSA)'.
565

**Author changes:** Thank you for your valuable comments and suggestions. We have revised the paper and made amendments.

570 and what possible drawbacks/bias have to be considered, i.e. Example 1: comparing annual values of accumulation rate – by averaging the annual accumulation rates of all the three firn cores (arithmetic average?), problem: lack of high-res density – what is the influence of intrusion, single events on the overall record/correlation analysis;

575 **Author response:** Reviewer 1 has asked for additional analysis on the influence of extreme precipitation events at MBS. We propose to use the data from the Modèle Atmosphérique Régional to look at the frequency of precipitation at MBS at a monthly/seasonal timescale. Please note that detailed investigation into the extreme precipitation events at MBS will be undertaken shortly, as this is part of the first authors (Camilla Crockart) PhD project. The annual accumulation rates and sea salt concentrations from the three individual MBS ice cores (Alpha, Charlie and Main) have been compared are in agreement (please see Figure 3 and 4 table inserts).

580 **Author changes:** We have added additional analysis using the surface mass balance variable from the MAR model, instead of RACMO. Also, Vance et al. (2016) looked at RACMO and showed that it underestimated surface mass balance in the MBS region. We have added analysis looking at the climatology, seasonal variability and extremes in surface mass balance at MBS. Please see Fig. 4, and L462 in Methods, L518 in Methods, and L701, 796, 871 in Discussion.

585

Example 2: looking over a time period of 40 years (in comparison to ERA5 etc)- which is by no means close to centennial. What is the significance of this correlation over this time period with respect to a future millennial record?

590 **Author response:** The lack of observational data in the southern high latitudes, in particular the Indian Ocean sector, means the satellite era is the only period of time when we can intensively explore mechanisms, transport and signal preservation in the MBS record. This is standard for high latitude/Antarctic studies. That this period may not properly represent centennial climate is clear (e.g., Jones et al., 2016). However, the development of high-resolution paleoclimate records, such as MBS and LD, is an effort to extend our understanding of variability in this region. In relation to the statistical significance of the correlations with climate modes, we expect these 595 correlations would only increase ($p$-value decrease) over a longer time period (i.e., with a greater n). Please note that we used effective degrees of freedom for all $p$-values presented in this paper to eliminate the influence of autocorrelation within any of the timeseries (L228). Future work will definitively explore the longer timeseries using longer datasets (e.g., longer ENSO and SAM indices once we have more MBS data dated and available).

600 **Author changes:** See above.

Did you test for trends (affecting the correlation?) What if the correlation is (only) a result of recent changes and not due to variability itself? For variability analysis the trends have to be looked at (and removed)?

605 **Author response:** Yes, we made sure to detrend all of the time series data (including ice core data, climate mode indices, ERA5 and the HadISST data). Therefore, the correlation results are not due to trends in the time series. L254 read 'The ENSO, SAM and IOD indices were detrended (as were the corresponding accumulation and sea salts time series) to reduce the interference of any climate change signals and ensure any significance was due to inter-annual variability, rather than (for example) the pronounced shift toward the positive SAM phase during 610 austral summer in recent decades (Marshall et al. 2003; Thompson and Wallace 2000)'.

**Author changes:** See above.

- a better structure in distinguishing between results and discussion –

615 **Author response:** Apologies. This will be carefully edited so that there is less overlap between the results and discussion. We propose to address these issues in detail in a marked-up version of the manuscript.

**Author changes:** This has been carefully edited so that there is less overlap between the results and discussion.

620

a focus on the available time scale, the high-resolution data available and what one can learn from it.

**Author response:** We will re-write sections of the abstract, introduction and conclusion to highlight the fact that we are only looking at the upper portion of the MBS record. This period also overlaps with the satellite era, meaning that we can use reanalysis data to investigate environmental conditions associated with any climate signals preserved in the MBS record. However, we think it is important to introduce in this paper – the first MBS data paper – the extended Main MBS record, as its length combined with its location is what makes this record unique for the East Antarctic region. We appreciate that this may be somewhat frustrating to the reader, however, there is extensive work in developing long ice core records, and we still think it valuable to publish an initial study of the upper (satellite era) section of the record that has thus far been developed.

**Author changes:** We have specified in the introduction that we are only looking at high-resolution record collected from the Indian Ocean sector of Antarctica (see L44). We have also emphasised that we are only looking at the upper section of the of the MBS record, and that the longer record is in development (see Abstract and Conclusion).

Further general remarks: 1. Terms on climate variability time scales are used inflationary and ambiguous throughout the text, for example: High-resolution ice core – long term climate variability and multi-centennial – all in one sentence/context. (Abstract, Line 21-22, Introduction Line 44), Usage of very different time scales (as relevant for the paper), for example: variability over past millennia (Intro, Line 55), sub-decadal (?) signals for climate variability (Intro, Line 126), multi-centennial (Intro, Line 86) Suggestion: There should be a clear outline of which times scales the author aim to address with their data – both in resolution (sub-annual) and coverage (30-40 years!).

**Author response:** This will be simplified, for example, we will change the introduction to 'Ice cores collected from the Antarctic ice sheet contain chemical signals that are used to reconstruct past climate conditions. Ice cores can be categorised as low-resolution (centennially-resolved) and high-resolution (annually-resolved) records. Low-resolution ice cores collected from low-accumulation zones of the Antarctic Plateau contain climate signals dating back hundreds of thousands of years. High-resolution ice cores contain more detailed climate signals but back date only millennia.'

**Author changes:** This has been revised, please see L20 and L38.

It is in the nature of coastal high-res records, that they resolve-sub-annually (a benefit which is unfortunately not discussed or presented here more in-depth). Overall the (only) time scale considered here is annual mean over a period of 40 years – for both accumulation rate and sea salt concentration.

**Author response:** Some high-resolution ice core records can be sub-annually resolved (e.g., the LD record, McMorrow et al. 2001; Vance et al., 2013; 2015). However, here we chose to use the annual MBS accumulation and sea salt records to avoid any errors associated with assuming a uniform accumulation at the site, which has not yet been tested definitively. We fully expect to develop sub-annual records in the future, but this requires extensive analysis on the extreme precipitation events at MBS, which is too large a task for the current manuscript.

**Author changes:** See above.

2. Talking about climate signals/climate variability at different time scales would require a profound analysis of the climate signal contained in the record (especially given the fact, that overall a short period of 40 years are considered. What is the common climate signal in the three cores? And what the common climate signal with the core from Law Dome? Are the 40 years long enough? What about variability vs mean/trend?

**Author response:** As above - all of the time series have been detrended, and therefore any climate change trends do not influence the correlation results. The three individual MBS ice cores do not have a trend over time ($r = 0.122, 0.086, -0.034, p = 0.466, 0.598, 0.873$ for Alpha, Charlie and Main, respectively). That this period may not

properly represent centennial climate is clear (e.g., Jones et al., 2016). However, there is a lack of observational data in the southern high latitudes, meaning that the satellite era is the only period of time when we can intensively explore mechanisms, transport and signal preservation in the MBS record. The purpose of developing the MBS and LD records is to extend our understanding of variability in this region. Correlations between the LD record and climate modes can be found in Table 1 - the LD record is significantly correlated with the ENSO indices. We chose to use the MBS site average (which is in good agreement with all three of the individual MBS ice cores, please see table insert in Figures 4), because we wish to maximise the overlap time period with the satellite data as the Main MBS ice core only dates back to 2007. Moreover, using multiple records allows for improved signal to noise ratio (Jones et al. 2016). Please note that three international laboratories are working on the MBS record, with new data currently being developed (e.g., dust, stable water isotopes, volcanic signals, extreme precipitation event signals and visible stratigraphy).

**Author changes:** See above.

3. Already in the introduction 14 different abbreviations are introduced (and used later in the text) – its hard to read and to follow your argumentation. Maybe it is possible to stick to few, relevant terms – as not all of the modes are used later or relevant for the paper (or one could stick to summarized, overarching terms of comparable modes).

**Author response:** Good suggestion. We propose to revise using only standard acronyms (e.g., MBS, LD, IOD, ENSO, and SAM).

**Author changes:** We have revised and using only standard acronyms (e.g., MBS, LD, IOD, ENSO, and SAM).

4. Information to the firn cores Do you analyze the core until the surface? How do you deal with the upper meter(s). How do you cut the core in the field, transportation, cutting in the lab.

**Author response:** Greater detail on the drilling process of the individual ice cores will be added to the methods section. The Main MBS ice core was drilled from 4 m depth using a Danish Hans Tausen intermediate drill, while the short MBS cores were drilled from the surface using the Kovacs system.

**Author changes:** We have added more detail, please see L147.

A table of the exact coordinates, the length/logging depth etc is missing, and then the obtained coverage in time

**Author response:** Figure 1 will be changed to include a map with all four ice cores drilled at MBS (Alpha, Bravo, Charlie and Main). The time coverage and maximum depth for each MBS ice core will be made clearer, along with the coordinates of the individual ice cores.

**Author changes:** For exact coordinates of each MBS ice core please see Fig. 1, and for their time coverage, please see L295.

There is inconsistency in the number of cores included in your analysis. In chapter 2.1 (methods) it reads, that from the three MBS firn cores, the Bravo core is not used in the study. That makes 2 shallow cores plus the upper part of the main core. i.e three cores from the MBS site. However, in line 133 it reads: The MBS record is unique in that it contains three short ice cores (20-25m) (. . .) in addition to the Main core.

**Author response:** There were four MBS ice cores collected (Alpha, Bravo, Charlie and Main) however, the Bravo core is not extensively analysed here, and is only used for confirmatory purposes during dating the records presented here. Section 2.1 reads 'The cores drilled include one long core, MBS1718 (hereafter termed "Main", depth 295 m), and three short cores, MBS1718-Alpha, MBS1718-Charlie, MBS1718-Bravo (hereafter "Alpha", "Bravo", and "Charlie", 20-25 m depth). The Bravo core was collected exclusively for persistent organic pollutant analysis so it will not be considered hereafter, although high resolution water stable isotope analyses from this

725     core were considered for confirmatory purposes during dating the records presented here'. We will change L133 to 'The MBS record is unique in that it contains multiple short ice cores (20-25m)' to avoid confusion.

**Author changes:** Please see above, and Fig. 1 for location of all of the MBS ice cores.

730     There is also unclear usage of the term: "record" – is this always meant as the stacked (averaged) record over the three (?) cores? This should be stated clearly.

**Author response:** This will be clarified throughout the paper, with the average accumulation and sea salt records from the three MBS ice cores referred to the 'MBS site record'.

735     **Author changes:** This has been revised to *'MBS site record'* throughout the paper.

Comments in detail: Line 21: wording in combination: high-res ice core vs long term, multi-centennial versus long-term

740     **Author response:** This will be changed to 'Ice cores collected from the Antarctic ice sheet contain chemical signals that are used to reconstruct past climate conditions. Ice cores can be categorised as low-resolution (centennially-resolved) and high-resolution (annually-resolved) records. Low-resolution ice cores collected from low-accumulation zones of the Antarctic Plateau contain climate signals dating back hundreds of thousands of years. High-resolution ice cores contain more detailed climate signals but back date only millennia'.

745     **Author changes:** This has been changed as above, please see L38.

Line 30: occurrence of moisture and aerosol intrusions is briefly touched in the discussion but not shown in the data

750     **Author response:** Apologies, this will be changed to 'Further study on this new site may help to determine whether this inverse relationship is due to distinct moisture intrusions.' The first piece of evidence to help with this analysis is the work of Udy et al. 2021 (cited in this work), which showed that the two ice core sites have differing moisture sources at the synoptic scale, and our results support this finding. This will be made clearer in the manuscript.

755     **Author changes:** This has been changed, please see L28.

Line 52: high- resolution records are required to fill spatial gaps. . . High-res records address temporal information, not spatial?

760     **Author response:** Both the brevity and sparse distribution (particularly in East Antarctica) of high-resolution ice cores limits our understanding of climate variability (Jones et al. 2016; Stenni et al. 2017; Vance et al. 2016). The only millennial length high-resolution ice cores in East Antarctica from Enderby to Wilkes land have been collected from the Law Dome region. Hence, the extended Main MBS record does fill a spatial and temporal gap
765     in high-resolution ice cores in East Antarctica.

**Author changes:** See above.

Line 141: please explain, what is meant by "wet deposition"?
770
**Author response:** This will be changed to 'wet-deposition (deposition via precipitation and scavenging by blown snow, Legrand and Mayewski 1997)'.

**Author changes:** Please see L139.

775

Line 151: The MBS record: It is not explained, how you derive the "record" here – what exactly is done? The record of the single accumulation rates or the stacked/merged?

> **Author response:** This will be clarified throughout the paper, with the average accumulation and sea salt records from the three MBS ice cores referred to the 'MBS site record'.

> **Author changes:** When referring to the average sea salt or accumulation records between the three cores analysed here, we have written the *'MBS site record'*.

Line 160-165: In the figures only 1 record for LD is shown – how did you combine the old and new records? Where they exactly the same (in the common period)? A little note on how the two records are combined is needed here

> **Author response:** There is only one LD record presented here, which is a composite of a new short ice core (1990-2015) drilled in 2016 and the longer DSS97 ice core (1975-1989) drilled in 1997 (e.g., Zhang et al. cp-2020-124). Please note that we have written 'The new record presented here covers the period 1989-2016' however, the two composites were joined at 1989/1990, meaning that the new record starts at 1990, our apologies – this will be corrected. This data is an improvement on the composite of numerous short cores presented in Vance et al. (2013). At the time of writing Vance et al., 2013, the longer DSS97 record was extended into the satellite era to 2009 using four short 8-10 m ice cores drilled in the years between 1997 and 2009. The composite of two cores presented here improves on this record, principally because at a high snowfall site like Law Dome (>1.5 metres snow per year) a short ice core of less than 10 metres will not span 2009 to 1997. However, in 2016, we drilled a 30.8 m core using an eclipse drill, which enabled us to span the entire period from 1990-2016 with one complete core. The two composites were compiled using visual analysis of the raw data of overlapping seasonal cycles in 1989/1990 of key dating analytes (principally $\delta^{18}O$, nssSO$_4^{2-}$, Na$^+$ and the ratio of SO$_4^{2-}$/Cl$^-$).

> **Author changes:** Please see L169.

Chapter 2.4. Deriving accumulation rates from empirical density model? Are there now bag mean densities obtained in the field/lab? As accumulation rate is one major result of the study and it is based on the empirical equation, it would be good to show the density data (modeled in comparison to bag means). In any case, different layers of density will not be considered and may bias the derived ice equivalents. A more in-depth description/uncertainty analysis should be given (again, based on the fact, that this is one of major results of the paper).Line 198-199: I do not understand: You convert your profile to ice equiva- lent in order to do exactly this: to compare different layers of different depth of an ice core, no matter of the thinning (by compaction). If you refer to thinning because of flow then it reads very strange, given the fact, that you are looking at 25m depth max.

> **Author response:** Yes, there are bag mean densities obtained in the field. These densities are essentially the same ($r = 0.957$, $0.956$, $0.975$, $p = 0.000$ for Alpha, Charlie and Main, respectively) as vertical thinning is negligible in the upper 20-25 m of the ice core. However, we choose to use the empirical density model to calculate accumulation so that future papers looking at the 1,000-year Main MBS record can directly compare snow accumulation rates.

> **Author changes:** Apologies, the empirical density model did not include vertical thinning. Bag densities were collected in the field (see L149). Please see L200.

Equation 1: description/ labeling of the terms is missing (i.e d = depth, what are the number standing for?)

> **Author response:** Good suggestion. Additional detail on each of the variables will be explained in more detail. In terms of the depth, L196 reads 'The empirical density model uses the mid-point depth of the annual layer (d)'.

> **Author changes:** Please see L201.

Line 280: The MBS record. . . Again, it is not clear to what is referred here.

830       **Author response:** This will be clarified throughout the paper, with the average accumulation and sea salt records from the three MBS ice cores referred to the 'MBS site record'.

      **Author changes:** As above, when referring to the average sea salt or accumulation records between the three cores analysed here, we have written the *'MBS site record'.*

835

Figure2: why do you not show the full record?

      **Author response:** We do not show a full figure as this would be too large, and it is common to use an example section (for example Sigl et al. 2016, see Figure 4 therein). Note that we have used a portion of the record that

840       shows not only years that were easier to discern, but also a period that was more difficult to date, in order to be upfront about the dating errors that may be incurred in the deeper record (see text in the marked-up version).

      **Author changes:** See above.

845 Line 302/303: wind/blown snow effect- where has this been shown? (Reference or short explanation how)

      **Author response:** The sensitivity test indicated that the annual total snowfall derived from ERA-5 correlated most highly with ice core total precipitation when the threshold for wind saltation was set very high (i.e., the nett effect of wind distribution is close to zero at this location). This will be added to the revised manuscript.

850

      **Author changes:** Please see L273 and L310.